**METHOD**

# Predicting the impact of sequence motifs on gene regulation using single-cell data

Jacob Hepkema[1], Nicholas Keone Lee[1,2], Benjamin J. Stewart[1,3,4], Siwat Ruangroengkulrith[5], Varodom Charoensawan[5,6,7], Menna R. Clatworthy[1,3,4] and Martin Hemberg[1,2,8*]

*Correspondence:
mhemberg@bwh.harvard.edu

[1] Wellcome Sanger Institute, Wellcome Genome Campus, Hinxton CB10 1SA, UK
[2] The Gurdon Institute, University of Cambridge, Tennis Court Road, Cambridge CB2 1QN, UK
[3] Molecular Immunity Unit, Department of Medicine, University of Cambridge, Cambridge CB2 0QQ, UK
[4] Cambridge University Hospitals NHS Foundation Trust and NIHR Cambridge Biomedical Research Centre, Cambridge CB2 0QQ, UK
[5] Department of Biochemistry, Faculty of Science, Mahidol University, Bangkok 10400, Thailand
[6] Integrative Computational BioScience (ICBS) Center, Mahidol University, Nakhon Pathom 7310, Thailand
[7] Systems Biology of Diseases Research Unit, Faculty of Science, Mahidol University, Bangkok 10400, Thailand
[8] Gene Lay Institute of Immunology and Inflammation, Brigham and Women's Hospital, Massachusetts General Hospital, and Harvard Medical School, Boston, MA 02115, USA

## Abstract

The binding of transcription factors at proximal promoters and distal enhancers is central to gene regulation. Identifying regulatory motifs and quantifying their impact on expression remains challenging. Using a convolutional neural network trained on single-cell data, we infer putative regulatory motifs and cell type-specific importance. Our model, scover, explains 29% of the variance in gene expression in multiple mouse tissues. Applying scover to distal enhancers identified using scATAC-seq from the developing human brain, we identify cell type-specific motif activities in distal enhancers. Scover can identify regulatory motifs and their importance from single-cell data where all parameters and outputs are easily interpretable.

## Background

One of the central goals of genomics is to understand how different phenotypes are determined by the DNA sequence. In higher eukaryotes, the vast majority of bases represent non-coding DNA, and our ability to predict the function of these sequences is incomplete. An important role for non-coding DNA is to regulate the expression of protein-coding genes, and a key mechanism is through the binding of TFs to proximal promoters and distal enhancers. Although the underlying principles and mechanisms of TF binding have been studied extensively, determining which motifs are functional and quantifying their importance remains a major challenge [1–3].

Identifying regulatory motifs from sequence alone is hard because in higher eukaryotes, most motifs are degenerate and the number of matches in the genome is typically much larger than the number of sites bound by a TF [4]. Using epigenetic information, such as TF expression levels, TF binding, chromatin accessibility, and histone modifications, it is possible to obtain more accurate models of which motifs are relevant for expression [5, 6]. Today, assays for quantifying the epigenome can be carried out for single cells [7–9], resulting in datasets with finer cellular resolution. Although there are some packages available for processing such data [10], additional methods that can take

full advantage of these data to infer regulatory motifs and their impact on molecular phenotypes, such as expression levels or open chromatin, are required.

Although there are methods for inferring regulatory motifs from single-cell data, e.g., SCENIC [11], HOMER [12], and Basset [13], none of these methods is designed for both de novo motif identification and quantification of the contribution to gene expression levels with scRNAseq. A variety of machine learning approaches have been applied to bulk RNA-seq and ChIP-seq data to identify sequence and epigenetic features that determine gene expression [6, 14–17]. In recent years, deep learning methods have become very popular [18, 19] in genomics. The premise and the specific task solved by these methods vary. One set of methods, e.g., BPNet [20], Enformer [21], and Basset [13, 22], require a signal (e.g., chromatin accessibility) to be associated with the sequence, and they can be used to predict the values for sequences that have not been observed during training. In doing so, these methods learn sequence motifs, but with a multi-layered architecture, the representation is distributed and may be difficult to interpret biologically, although methods such as BindVAE [23] have been successful in using other approaches. This setup, however, is not ideal for predicting gene expression, so different approaches have been proposed [6, 24, 25]. A more detailed comparison to other deep learning approaches can be found in Additional file 1: Table S1. To the best of our knowledge, there are no methods available for simultaneously discovering regulatory motifs and determining their contribution to gene expression using single-cell data. Thus, we developed scover to achieve these tasks. We envision that scover will be a useful tool for researchers interested in determining which motifs are most important for the observed signal from a single-cell experiment. This cell type-specific, quantitative information is likely to be helpful when trying to extract a shortlist of motifs and TFs that are most important, e.g., for low throughput validation experiments.

## Results

Here, we perform de novo discovery of regulatory motifs and their cell lineage-specific impact on gene expression and chromatin accessibility from single-cell data using scover, a convolutional neural network (Fig. 1a, b). Scover takes as input a set of one-hot encoded sequences, e.g., promoters or distal enhancers, along with measurements of their activity, e.g., expression levels of the associated genes or accessibility levels of the enhancers. The output is a set of motifs associated with the convolutional filters along with a vector of influence scores for each motif in each output pool. Scover was implemented using the PyTorch framework [26] to be compatible with the scanpy workflow [27], and it is available under the MIT license at https://github.com/jacobhepkema/scover.

Single-cell measurements from sequencing experiments typically have a large fraction of zeros. To overcome the accompanying challenges [28], scover reduces the sparsity by summing the values for $k$-nearest neighbors (default $k=100$) of a set of seed cells to generate a "pooled" dataset (Fig. 1a). This strategy does not rely on existing cell type annotation, and it also retains intra-cell type variability, as opposed to taking the average for existing cell annotations. The initial seed cells are selected by geometric sketching, which evenly samples cells across a representation of the dataset, preserving rare cell states [29]. The use of the $k$-nearest neighbor graph ensures that most cells in a pool will

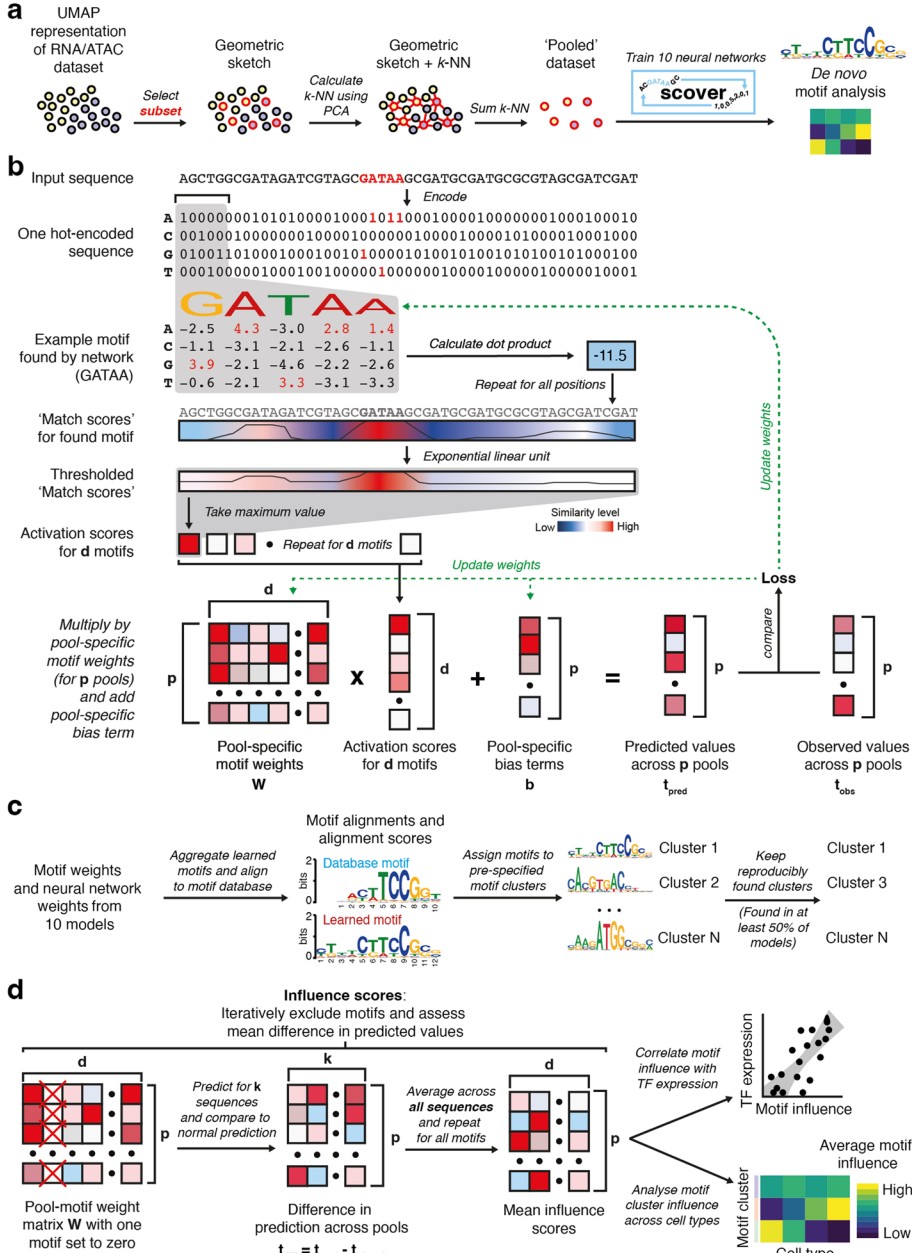

**Fig. 1** Overview of the scover workflow. **a** Cell pooling strategy. A subset of cells is selected using geometric sketching. The *k*-nearest neighbors for cells in the subset are summed to produce a less sparse representation of the original dataset. **b** Scover infers regulatory motifs that are predictive of the signal associated with a set of sequences using a neural network consisting of a single convolutional layer, an exponential linear unit, global max pooling, and a linear layer with bias term. **c** The identified motifs are merged and compared to annotated motifs and assigned to pre-specified motif clusters based on their most significant alignment. **d** To evaluate the contribution of each motif, an influence score is calculated using a leave-one-out strategy

come from the same cell type (Additional file 2: Fig. S1a). The cell type annotation for a pool is determined by the most abundant cell type within the pool.

We use several metrics to evaluate the effect of different pool sizes (Additional file 2: Fig. S2). Reassuringly, we have found that results for the datasets considered in this

manuscript are robust with respect to the pool size, and it is up to the user to strike the balance between model accuracy and cell type resolution. We also found that the pools tend to be relatively homogenous, with one cell type accounting for $> 80\%$ (Additional file 2: Fig. S1). We confirmed that an alternative pooling strategy [30] resulted in similar homogeneity, suggesting that our approach is robust. Sequences are fed into a single convolutional layer where $d$ (default $d = 600$) filters representing regulatory motifs are combined before being fed to a neural network layer resulting in a predicted expression value. The network includes a bias term which is independent of the input sequence, and it can be thought of as representing all other determinants of gene expression. The shallow architecture requires fewer parameters which allows smaller input datasets to be used, and it makes it more likely that a whole motif representation will be found [31].

Since the neural network is optimized using a stochastic approach, reproducibility can be achieved by running it $r$ times (default $r = 10$) and excluding motifs that are found in fewer than 50% of models [32]. To facilitate the overview of the found motifs, scover automatically compares the motifs to those annotated in CIS-BP [33] (if available). Randomly initialized convolutional filters may converge to recognize similar sequences that are enriched in the sequence input. To organize the $rd$ motifs, they are assigned, based on their most significant motif alignment, to pre-specified motif clusters published elsewhere [34] (Fig. 1c).

Analysis of the convolutional filters and their associated weights can provide insights into the contribution of regulatory motifs to the observed measurements. To evaluate the importance of each motif, we employed a strategy inspired by Maslova et al. [32] whereby an influence score $n_{ip}$ is calculated for each motif $i$ and pool $p$ by excluding the motif from the analysis and comparing the prediction of the modified model to the original model (Fig. 1d). Influence scores in pools corresponding to the same cell type are averaged and influence scores for motifs belonging to the same motif cluster are summed. Although influence scores vary across pools, the set of motifs represented by the convolutional filters is the same for all pools. Since influence scores tend to be largely similar across cell types, we often present the $z$-transformed values to highlight cell type-specific patterns.

When multiple TFs recognize the motif of a motif cluster, additional information is required to identify the most likely protein. Although transcriptome data alone is insufficient, we can rank the candidates by calculating the Spearman correlation $r_{ct}$ between the summed influence scores for motifs in motif cluster $c$ and the expression levels of TF $t$ across all pools. We assume that if $|r_{ct}|$ is high, then the TF is more likely to bind to its cognate site.

### Scover identifies regulatory motifs in the human kidney

We applied scover to a scRNAseq dataset from human fetal and adult kidney containing a total of 67,471 cells [9]. The 1-kb input sequences were 500 nt upstream and downstream of the transcription start site (TSS) of each expressed gene. We used $d = 600$, and of the 6000 convolutional filters, 13.3% could be grouped into 16 reproducible families corresponding to known motifs (Fig. 2a, b, Additional file 2: Fig. S3), many of which have been previously reported to be important for the kidney based on analysis of chromatin

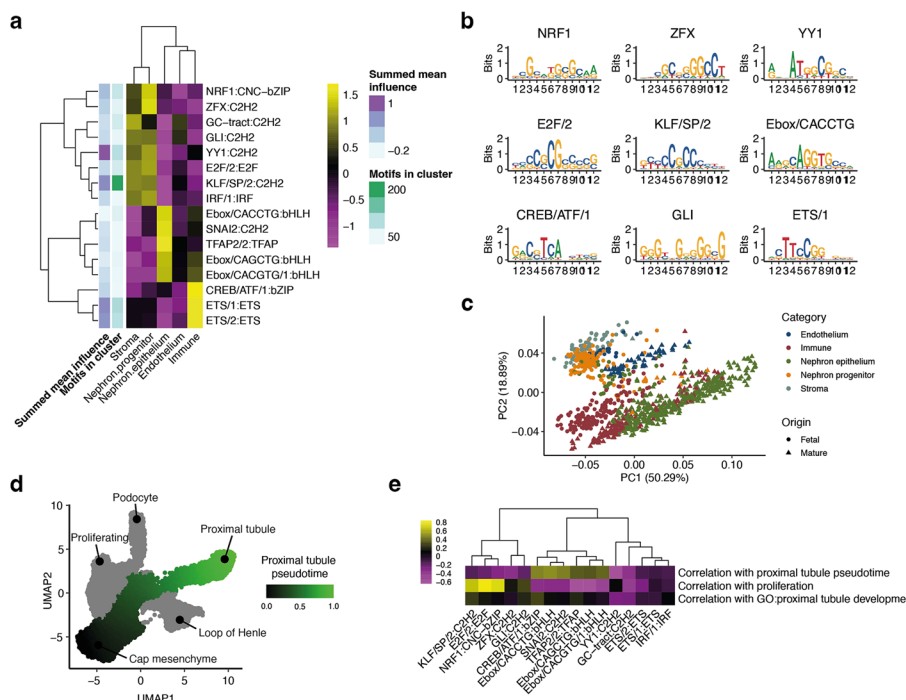

**Fig. 2** Analysis of proximal promoters for the fetal and adult human kidney. **a** *Z*-transformed influence scores for sixteen motif clusters and five cell categories. **b** Example motif logos from the sixteen motif clusters. **c** Projection of the pools onto the first two principal components of the influence score matrix reveals similar regulatory profiles. **d** Pseudotime trajectory for nephron progenitor development across single cells in fetal kidney represented as a UMAP plot. **e** Spearman correlations between motif influence scores and mean pseudotime values or expression of genes related to proliferation and proximal tubule development

accessibility experiments carried out in bulk [35]. Analysis of the top non-aligned motifs shows that many correspond to ETS motifs or GC-rich motifs, suggesting that they are likely biologically meaningful nonetheless (Additional file 2: Fig. S3). Based on the validation set, the scover model explains, on average, 15% of the variance in gene expression, and the performance drops to 9% on average if non-matching motifs are excluded (Additional file 2: Fig. S4). After permuting the order of the nucleotides in each sequence, the performance drops to 5% on average. To ensure that scover learns a cell type-specific model, we also carried out a control experiment whereby the pool order was permuted, after which only 6% of the variance could be explained on average. As an additional benchmark, we used FIMO [36] to first identify motifs, and then, we used either a linear regression or a random forest model with 20 estimators with the motifs as explanatory variables to predict gene expression. The two models could only explain 12% and 10% of the variance, suggesting that our strategy of simultaneously identifying motifs and their weights provides a better fit (Additional file 2: Fig. S4).

To simplify the presentation of the results, we grouped the 60 cell types into five categories: endothelial, immune, nephron epithelial, nephron progenitor, and stromal (Additional file 1: Table S2). To visualize the motifs, we also applied dimensionality reduction using principal component analysis (PCA) revealing two major groups, the first consisting of nephron epithelium and immune cells and the second consisting of nephron progenitors, endothelium, and stromal cells (Fig. 2c, Additional file 2: Fig. S5). The third

principal component separates immune cells and endothelial cells from the rest. This result is not surprising since the motif scores reflect the underlying expression matrix, and it suggests that the representation has not distorted the clusters.

Since one of the largest differences in terms of influence scores is between nephron progenitors and nephron epithelium cells, we investigated the developmental trajectory by carrying out a pseudotime analysis. This revealed how the progenitors branch towards three distinct fates: podocytes, loop of Henle, and proximal tubules (Fig. 2d). Focusing on the latter, we calculated the correlation between the expression levels of six markers associated with tubule development and the influence scores to reveal strong associations for SNAI2 and Ebox/CACCTG (Fig. 2e). This suggests that we can relate influence scores not just to discrete categories but also to continuous processes.

Although promoters share some distinct sequence features, there is also a considerable diversity to allow for differential regulation. To visualize this diversity, we applied UMAP dimensionality reduction to the matrix containing the inferred motif occurrences in each promoter, and then we assigned a color to each dot based on its motif score (Fig. 3a, the "Methods" section). Interestingly, promoters separated in terms of their motif contents; Ebox motifs were abundant across almost all promoters, while E2F and YY1 showed opposite enrichment. This indicates that some motifs are found at

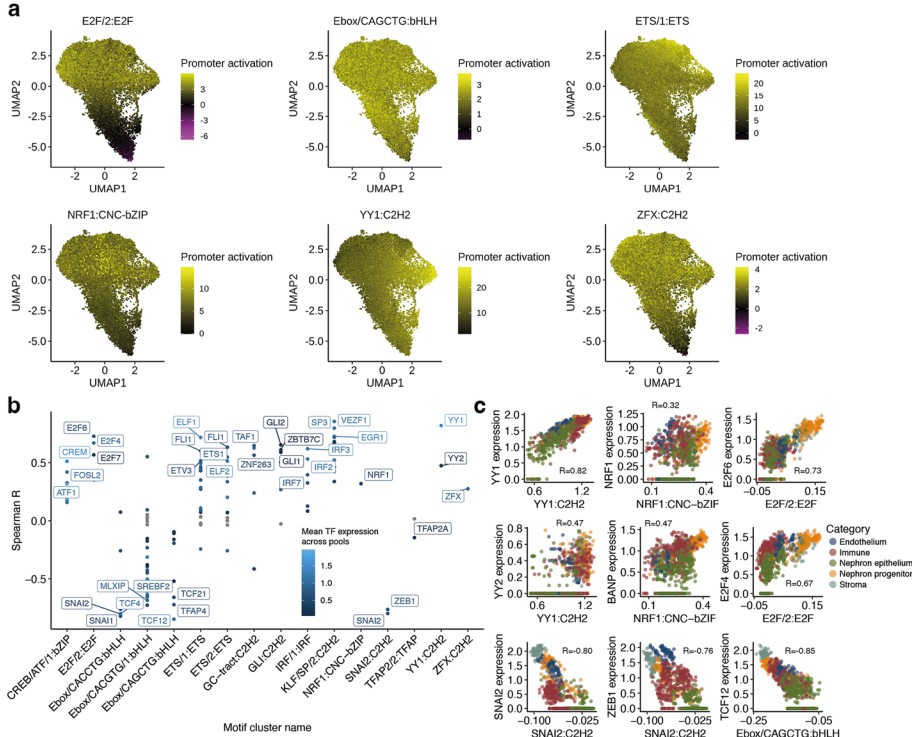

**Fig. 3** Motif cluster impact for the human kidney dataset. **a** UMAP plots of 18,150 promoters based on motif occurrences after the ELU and max-pool operations. Subplots show the summed max-pooled motif scores after the ELU and max-pool operations (promoter activation) for selected motif clusters in each promoter. **b** Spearman correlation between motif influence scores and expression levels of TFs binding to the motif in question. TFs with *p*-value ≥ 0.05 (after Benjamini-Hochberg correction) are in gray, the significant ones are colored by expression level, and selected top TFs for each cluster are named. **c** Correlation between motif influence and expression of selected TFs across pools. The Spearman correlation is reported for each panel

similarly regulated promoters, while others are found across a wide range of promoters and are thus likely to be involved in a wide range of programs. Similarly, by calculating the correlation between motif family influence scores across pools, we observe that the motif families fall into two major groups with GC-rich KLF and E2F motifs in one cluster, further reinforcing the notion that there is a higher-order organization of motifs (Additional file 2: Fig. S6).

Since multiple TFs can compete to bind similar motifs, we set out a strategy to identify which TFs could putatively underlie the observed influence scores for each motif family. We do this by correlating motif family influence scores with the expression levels from TFs in those motif families, under the assumption that TF expression relates to its activity, as has been suggested for NF-κB [37]. Comparison of TF expression and influence scores reveals that 55 of the 132 TFs associated with the 16 motif families are strongly correlated with $|r| > 0.5$ and a significant *p*-value (Fig. 3b). These correlations can be used to pinpoint which putative TFs more likely correspond to the network predictions. For instance, while *YY1* and *YY2* have very similar motifs with an ATGG core [38], *YY1* has a higher correlation, suggesting that it is less likely that *YY2* corresponds to the network prediction (Fig. 3c). Similarly, while *NRF1* has a moderate correlation with influence scores, it more likely corresponds to the recently characterized TF BANP that recognizes a similar CGCG motif [39], as it has a higher correlation with influence scores. However, we cannot rule out that multiple TFs correspond to the same motif cluster. For instance, *E2F4* and *E2F6* both correlate moderately with E2F/2 cluster network influence scores. A negative correlation between a motif cluster and a TF could suggest a repressive role. For instance, we find that the SNAI2 weights are strongly negatively correlated to SNAI2 expression, confirming its repressive role [40].

### Characterization of regulatory motifs in 20 mouse tissues

Next, we applied scover to 20 tissues from the Tabula Muris [8], containing a total of 67 cell types and 38,080 cells. Only 717 of the 6000 convolutional filters matched 13 different motif families, 11 of which were the same as for the human kidney (Fig. 4a, b, Additional file 2: Fig. S7).

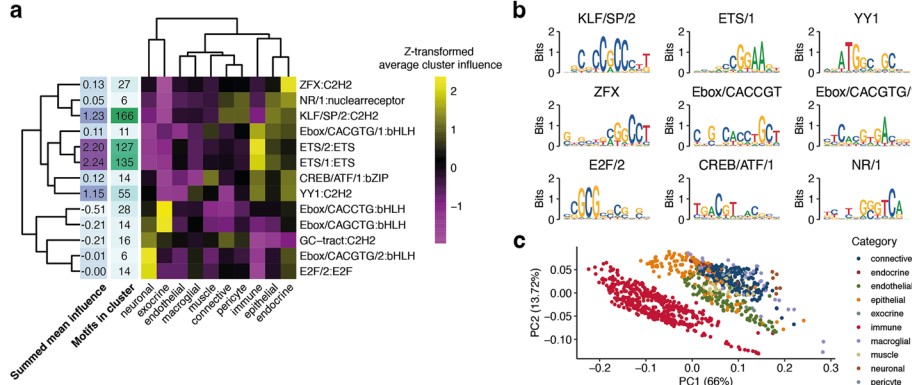

**Fig. 4** Analysis of proximal promoters from the Tabula Muris. **a** Z-transformed neural network influence for aggregated motif clusters. **b** Example motif logos from the thirteen motif clusters. **c** Pools are projected onto the space represented by the first two principal components of the motif influence matrix

We found that motifs that did not match any database motifs were very GC-rich and sometimes resembled known motifs (Additional file 2: Fig. S7), and in their absence, the fraction of variance explained is reduced from 28 to 14% on average. Moreover, permuting the order of the nucleotides in the sequences reduced the fraction of variance explained to 7%, and permuting the order of the pools reduced the fraction of variance explained to 14% (Additional file 2: Fig. S4). Similarly to the kidney data, using FIMO to find motifs, followed by linear regression or a random forest (with 20 estimators) to identify their weights provided a poorer fit, explaining 24% and 20% of the variance, respectively.

We categorized cell types as adipose, connective, endocrine, endothelial, epithelial, exocrine, immune, macroglia, muscle, neuron, and pericyte (Additional file 1: Table S3). Dimensionality reduction of the influence score matrix reveals three distinct groups: immune cells, neurons, and the remaining cells (Fig. 4c), consistent with the expression analysis [41]. The highest influence scores are found for ETS, followed by KLF and YY1 (Additional file 2: Fig. S8). We analyzed all promoters in the dataset for their motif composition (the "Methods" section). ETS was the most abundant motif, whereas KLF, E2F, and Ebox were specifically found in subsets of promoters (Fig. 5a, Additional file 2: Fig. S9).

We hypothesized that in addition to having higher activity in specific cell types, the identified motifs would also be associated with distinct sets of cellular processes. We obtained 16,610 gene lists representing a diverse set of processes from the Gene Ontology (GO) database [42], and for each list, we calculated the correlation between the average expression of the genes and influence scores across pools. We kept the terms with the top 1% of correlation scores, and we found that the motifs segregated into two main clusters, while the GO terms fall into three groups (Fig. 5b). Reassuringly, the influence scores for the motifs that are highest in immune cells were positively correlated with genes associated with immune-related terms. Some of the TF clusters in this group (ETS, YY1) have previously been reported to be important for immune cells [14, 32, 43–46]. The second group corresponds to GO terms that are primarily related to differentiation and morphogenesis. This group is characterized by high activity of E2F and KLF. The third group is more difficult to characterize, and we are unsure about the biological interpretation.

Comparison of TF expression levels and motif activity scores reveals several known regulatory principles (Fig. 5c). For example, the TFs with the highest correlation with ETS/1 influence scores are Elf1, Elf4, and Fli1. The high scores for ETS/1 in immune cells suggest that these TFs play a role in immune cell function. Indeed, Elf1 [47], Elf4 [48], and Fli1 [49] have all been described to play a role in regulating transcriptional programs related to the immune system. The negative correlation of Ebox/CACCTG with Tcf4 reflects its proposed role as a repressor [50]. In other cases, expression level can distinguish between TF influences: both Yy1 and Yy2 are similarly correlated to YY1 family influence levels, and they bind similar motifs [38], but Yy1 has higher expression levels, making it more likely to be the TF with a higher influence on downstream expression.

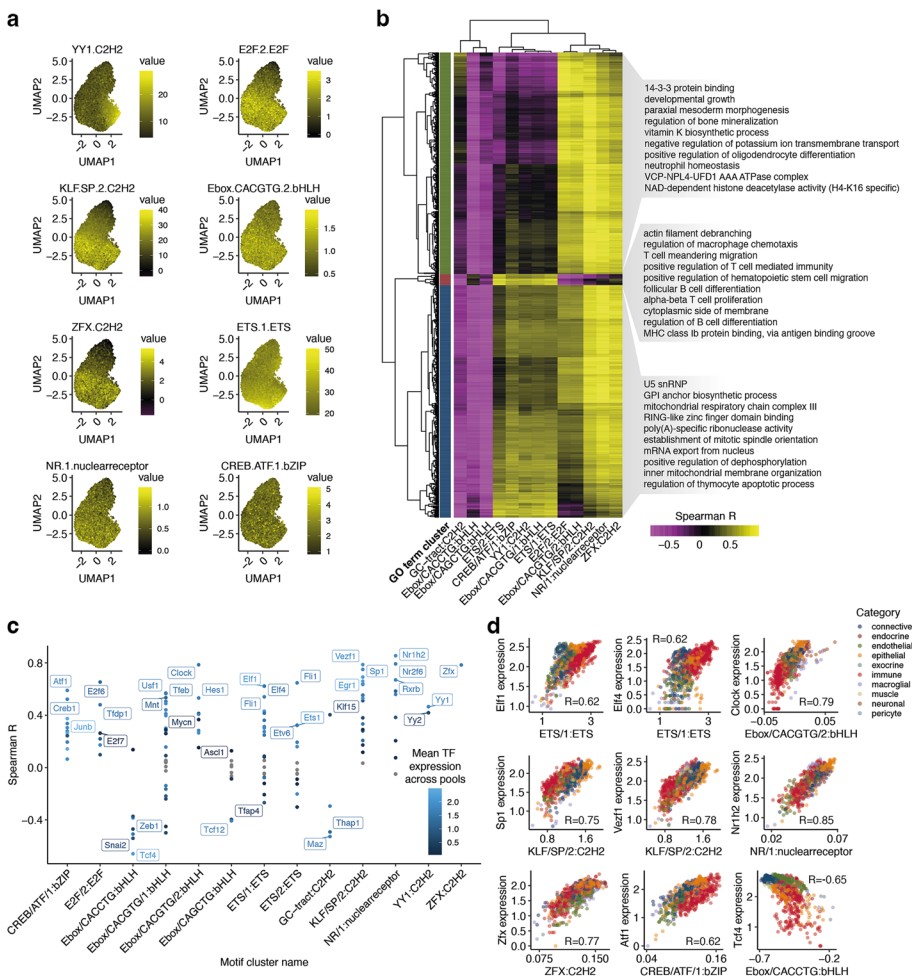

**Fig. 5** Motif cluster impact for the Tabula Muris dataset. **a** UMAP plots of 17,270 mouse promoters based on motif scores after the exponential linear unit and max pooling. Subplots show the summed motif scores (after the exponential linear unit and max pooling) for selected motif clusters in each promoter. **b** Spearman correlations between the expression of GO terms and aggregate motif influences in pools. Randomly selected enriched GO terms are shown for each cluster. **c** Spearman correlation between motif influence and expression levels of TFs binding to the motif in question. TFs with *p*-value $\geq$ 0.05 (after Benjamini-Hochberg correction) are in gray, the significant ones are colored by expression level, and selected TFs with the highest correlations for each cluster are named. **d** Correlations between aggregate motif family influence scores in pools and expression for selected TFs. The Spearman correlation is reported for each panel

## Identification of distal regulatory motifs in the human cerebral cortex

To demonstrate the versatility of scover beyond analyzing promoters and scRNAseq data, we used it to infer regulatory motifs from open chromatin by analyzing multimodal scRNA+ATAC-seq data from the developing human cerebral cortex [20]. Since we were interested in distal regulatory elements, only 292,338 loci that were outside the [−8 kb, 2 kb] region relative to annotated TSSs were considered. After training scover on the continuous accessibility of the open chromatin regions, we can relate the motif family influence scores back to candidate TFs by considering the expression values of the TFs in the cell pools (Fig. 6a).

After training on the accessibility data, we found 38 motif clusters (Additional file 2: Figs. S10 and S11). The fraction of variance explained by the model is similar to the

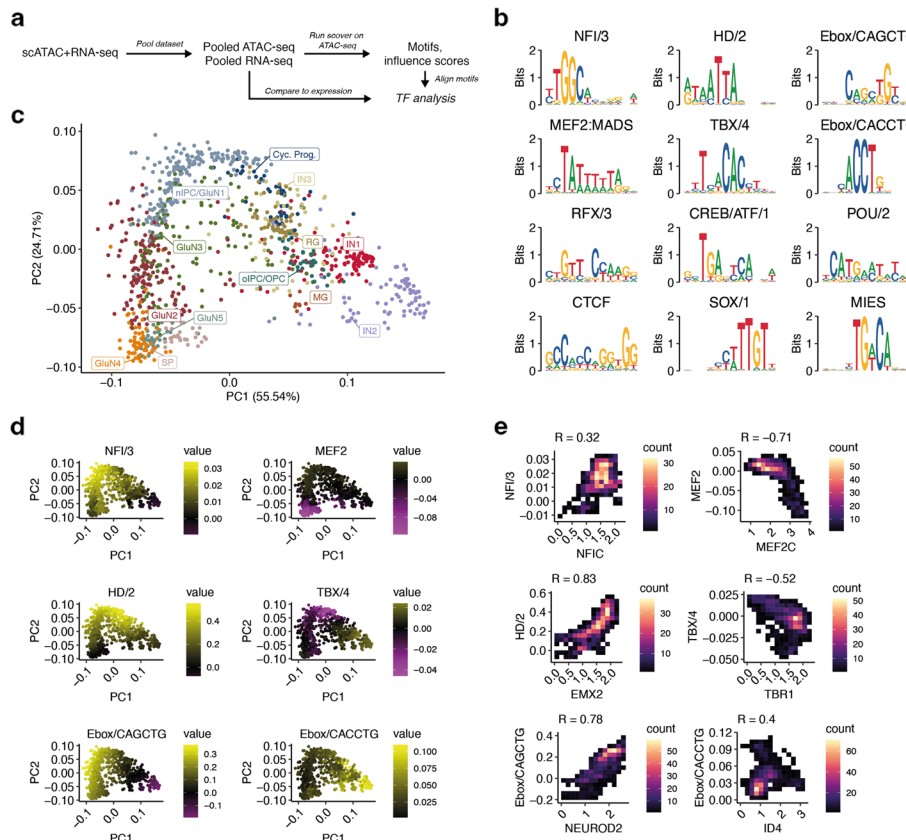

**Fig. 6** Analysis of distal open chromatin peaks from the developing human cerebral cortex. **a** Workflow for training scover on multimodal single-cell data. **b** Representative motifs of different motif families. **c** PCA plot fit on the motif influence scores across cells. **d** PCA plot as in **d** but showing the aggregate motif family influence scores of six example motif clusters across the pools. **e** 2D histograms showing TF expression in cell pools (*x*-axis) versus aggregate motif family influence scores in cell pools (*y*-axis)

kidney scRNAseq data with 19% (Additional file 2: Fig. S4). Again, the model is able to learn cell type-specific motifs as the fraction of variance explained is reduced to 8% upon permutation of the pools. Unlike the scRNAseq datasets, a linear regression model using motifs identified by FIMO performed slightly better, explaining 26% of the variance, suggesting that there is less of an advantage in jointly discovering motifs and their influence scores (Additional file 2: Fig. S4). We find that the most common motifs are HD/2 homeodomains, E-box, and RFX3 motifs. They also show the highest motif scores. We found more motifs than for the two scRNAseq datasets, suggesting a greater diversity than for promoters although it is worth keeping in mind that there were almost an order of magnitude more distal regulatory elements here. Similar to the scRNAseq data, there is a range of almost two orders of magnitude in the influence scores of the different clusters (Additional file 2: Fig. S12).

A low dimensional representation of the influence scores again reveals that pools corresponding to the same cell type are grouped together, with closely related cell types, e.g., the glutamatergic neurons, in close proximity (Fig. 6c). By overlaying the motif influence scores, we note that several motifs exhibit differential activity across cell types (Fig. 6d). Given the large number of motif combinations that are possible, this implies

that chromatin accessibility may be tuned in a cell type-specific manner. Interestingly, the two most common and influential motifs, two E-boxes recognizing CACCTG and CAGCTG, show opposite patterns of their influence scores, despite subtle differences at the sequence level.

Finally, we used the associated scRNAseq data to investigate the expression levels of the corresponding TFs, providing further insights regarding several motifs and TFs (Fig. 6e). For example, MEF2 genes have been reported to act both as activators and repressors depending upon their co-factors [51]. Our analyses suggest that for this phase of development, MEF2 motifs are associated with smaller peak sizes in glutamatergic neurons, but with no significant effect in other cell types. Moreover, comparison with expression levels strongly indicates that MEF2C reduces the chromatin accessibility for distal loci. Similarly, we observe a repressive function of TBR1, consistent with previous reports [52].

## Discussion

Here, we have presented a convolutional neural network model, which can be applied to a set of sequences, e.g., a subset of promoters or open chromatin regions, each associated with a vector of real values. Thus, the model is flexible and can be applied to a wide range of scenarios. In particular, it is well suited for single-cell data, and here, we present an exploration of inferred regulatory motifs and their activities across cell types using two different modalities of single-cell data. Previous attempts to develop quantitative models of regulatory motifs have been restricted to tissue level or cell lines [6, 14] or to immune cells [32, 43], and they frequently use histone marks rather than sequences to predict expression levels [53]. Unlike existing motif analysis methods which mainly identify overrepresented motifs in sequences [35], scover also determines the relative impact of each motif on gene expression. By comparing to gene expression levels, these scores can be used to pinpoint putative TFs that are related to the observed pattern.

Since we used a de novo strategy for finding motifs, we expect that it will be especially helpful for researchers studying organisms where regulatory motifs are poorly annotated. For mouse and humans, non-matching motifs make a meaningful contribution, but from our results, it is not yet clear what they represent. Some of them are likely good matches that have failed to converge, whereas others pick out CpG dinucleotides which are known to be associated with higher expression at promoters [6]. One shortcoming of scover is that only a minority of the motifs end up being used by the model, and this is computationally costly. By contrast, other methods such as BindVAE [23] have a more efficient motif usage model, and we believe that exploring better motif representations is an important future direction of research. Our results are also consistent with other studies that have categorized TFs as either preferentially binding promoters or enhancers [43]. One such example is HNF4A, which is important in the kidney [54], but most likely due to its preference for distal enhancers, it was not identified by our method [55].

Interestingly, there are several trends that are common to the kidney and the Tabula Muris datasets. Immune cells have regulatory profiles distinct from other cell types, and they have the highest influence on ETS motifs, while KLF and ZFX have a low impact (Figs. 2a and 4a). More generally, for the eleven motif clusters that are found in both datasets, the influence scores are in good agreement with the highest impact for the combined

ETS families, YY1, and KLF. A notable difference is that the quality of the fit provided by the Tabula Muris is substantially better. Since the two datasets have a similar number of cells and similar sparsity after pooling, we conjecture that this is due to the greater diversity of cell types in the Tabula Muris and the use of different experimental platforms.

Our finding that motif influence scores are related to genes associated with specific pathways findings (Fig. 5b) is indicative of higher-order organizational principles determining the motif arrangement in promoter regions. This is consistent with previous studies which have demonstrated that regulatory motif occurrences at promoters are associated with specific GO terms [36, 56]. Similarly, cross-species comparisons have shown that complex behavioral phenotypes are associated with specific transcription factor motifs [57, 58]. The underlying mechanism is most likely the 3D organization of the chromatin [59].

Importantly, the scover framework is not limited to gene expression, and we demonstrated how it can be used to predict chromatin accessibility. Interestingly, a simple regression model based on known motifs performed better, suggesting that more complex neural network models or more training data are required for this problem. It is also worth noting that there is room for improvement as neither scover nor the regression model took advantage of the gene expression data. Instead, the expression data was used in the post hoc analysis to relate TFs to motifs.

Although larger and more deeply sequenced datasets could allow for the discovery of additional motif clusters and better model fits, there are inherent limitations to the accuracy by which gene expression can be predicted from endogenous sequences. First, studies of gene expression noise have estimated that some of the variability is inevitable due to random fluctuations [60]. Second, there are many other mechanisms of gene regulation that are not included in proximal sequence data, e.g., transcript sequence features that influence mRNA stability, TF concentrations, distal enhancers, histones, miRNAs, and RNA-binding proteins. Third, we have only used one promoter for each gene, even though it is well known that many genes have alternative TSSs which could drastically impact regulation. Nevertheless, our results are consistent with a recent study of mouse hematopoietic cells which used epigenetic data to estimate that ~50% of the variance in expression can be explained by the endogenous promoter [53].

## Conclusions

We have demonstrated that a convolutional neural network can be used to simultaneously infer regulatory motif sequences and their relative importance from single-cell data. Using scRNAseq data, this allows us to find TF-binding motifs at promoters along with their impact on gene expression levels. Application to data from human kidneys and multiple mouse tissues revealed common regulatory patterns. Finally, we demonstrated that the framework can also be used for scRNA+ATAC-seq data and analyses of data from the developing human brain.

## Methods

### Pooling

Scover was trained as a regression model to predict the log-transformed pooled gene expression values (a vector of size $p$) from the associated promoter sequence of the gene. To reduce the sparsity, pools are created by grouping $k$-nearest neighbors of an

initial selection of cells ("seed cells") calculated using a shared UMAP embedding into $p$ equally sized groups of $q$ cells (Fig. 1a).

The following python workflow was followed for constructing the datasets. For the human kidney dataset, the fetal and adult kidney datasets were concatenated using the *concatenate()* function of AnnData objects. Then, using the command *sc.tl.pca(ad, svd_solver='arpack', n_comps=100)* from Scanpy, the first 100 principal components were calculated. For all datasets, the first PC was not used for subsequent steps since it correlated highly with the number of counts in the pools. We removed it by running *ad.obsm['X_pca']=ad.obsm['X_pca'][:,1:].* As the RNA datasets consist of concatenated datasets of different batches, batch-balanced $k$-nearest neighbor algorithm BBKNN was run with default parameters using the command *bbknn(ad, batch_key='dataset')* [61]. Subsequently, these $k$-nearest neighbors were used to calculate the UMAP representation using the Scanpy command *sc.tl.umap(ad).* Using geometric sketching, a subset of 1000 cells (the "seed cells") was selected that spans the UMAP representation of the dataset uniformly [29]. This was done using the geometric sketching function *gs()* from the "geosketch" package: *sketch_index=gs(ad.obsm['X_umap'], 1000, replace=False).* Then, $k$-nearest neighbors of these initial "seed cells" were calculated again, but now, $k$ was set to be the pool size. For the implementation of this method, please refer to the function *pool_anndata()* in the GitHub file *scover/data/utils.py* at https://github.com/jacobhepkema/scover. Briefly, $k$-nearest neighbors were calculated using the Scanpy command *sc.pp.neighbors(ad, n_pcs=40, n_neighbors=k),* where $k$ is the pool size. For each of the seed cells, the pooled counts were calculated as the sum of the raw counts for the seed cell and its $k$-nearest neighbors, followed by a log(1+counts) transformation.

For the Tabula Muris dataset, the workflow was similar. Twenty Tabula Muris FACS-sorted Smart-Seq2 datasets were concatenated in the cell dimension using the *concatenate()* function of AnnData objects. A total of 297 erythrocytes were excluded from further analysis given their low counts. The datasets included were Trachea, Bladder, Heart, Limb_Muscle, Diaphragm, Fat, Lung, Aorta, Mammary_Gland, Brain_Non-Myeloid, Skin, Kidney, Liver, Pancreas, Tongue, Brain_Myeloid, Thymus, Spleen, Marrow, and Large_Intestine. PCA was calculated using the same command as for the human kidney dataset, and the same command was used to remove the first PC. BBKNN was run using a slightly different command: *bbknn(ad, batch_key='dataset', trim=10,000, approx=False, use_faiss=False),* since this command was also used by the BBKNN developers for their integration of the Tabula Muris datasets. Then, the UMAP representation was calculated using the Scanpy command *sc.tl.umap(ad, min_dist=0.3).* The seed cells were obtained using the same command as for the human kidney dataset. The function *pool_anndata()* from our package was used with *neighbors=80* to create pools that aggregate 80 nearest neighbors each. Before training, the human kidney and Tabula Muris datasets were further filtered in a dataset-specific way as described in the "Data processing" section.

A second task we applied Scover to was to predict the log-transformed pooled chromatin accessibility values (again, a vector of size $p$) from the associated accessible region sequences. For this, we used a multimodal scRNA+ATAC-seq dataset of the human brain, so that we can compare learned motif influence scores to the transcription factor expression. For the human brain dataset, we calculated the first 100 PCs for

the RNA modality using the Scanpy command *sc.tl.pca(ad_rna, svd_solver='arpack', n_comps=100)*. Afterwards, the first PC was excluded using the command *ad_rna.obsm['X_pca']=ad_rna.obsm['X_pca'][:,1:]*. Since there were not multiple datasets to be integrated, BBKNN was not applied. Instead, nearest neighbors were calculated using the Scanpy command *sc.pp.neighbors(ad_rna, n_neighbors=100, n_pcs=80)*, and the UMAP representation was then calculated using Scanpy command *sc.tl.umap(ad_rna)*. The seed cells were obtained using the command *sketch_index=gs(ad_rna.obsm['X_umap'], 1000, replace=False)*. Next, the RNA dataset was pooled using the *pool_anndata()* command from our GitHub package with argument *neighbors=100* to create pools that aggregate 100 nearest neighbors each. Then, the same cells that were used to pool the RNA modality were used to pool the ATAC modality of the dataset using the *pool_anndata_given_pseudobulk_idx()* function from our GitHub: *pool_anndata_given_pseudobulk_idx(ad_atac, pseudobulk_idx)*. Before training, the dataset was further filtered in a dataset-specific way as described in the "Data processing" section.

### Convolutional neural network architecture

Scover is a convolutional neural network composed of a convolutional layer, an exponential linear unit (ELU) activation layer, a global maximum pooling layer, and a fully connected layer with bias and multiple output channels. The convolutional layer takes as input the one-hot encoded DNA sequences and the fully connected layer outputs predictions across the pools. Sequentially, the neural network carries out the following operations:

1. 2D convolution without bias values with $d$ convolutional filters of size $(m, 4)$
2. An ELU activation layer
3. A global maximum pooling layer that takes the maximum value for each of $d$ outputs of step 2
4. A fully connected layer with bias values with $d$ inputs and $p$ outputs

The number of output pools $p$ depends on the number of cells per pool $q$. The default values for running the model are $d=600$, $m=12$, and $q=100$. However, it is recommended that the user explore different combinations.

### Training the neural network

The neural network is trained in two stages: Bayesian hyperparameter optimization and a training stage using the best hyperparameters.

In the first stage, the hyperparameters are optimized through Bayesian hyperparameter optimization [62, 63]. The hyperparameter search is implemented using Ray Tune and hyperopt and runs distributed across available GPUs. For hyperparameter tuning, we use the ASHA algorithm, which applies aggressive early stopping to trials that do not seem to produce good results, finding optimal hyperparameters faster than random search.

The division of the dataset into training, test, and validation sets was performed using cross-validation. The dataset was divided into $K$ folds (default: 10) to create the "outer test" set (10% of data each) and the "inner" sets (90% of data each). Each

"inner" set was consequently split into a training and a validation set of 80% and 20% of the "inner" set, respectively. The "inner" sets were used for the hyperparameter search for each outer fold.

The hyperparameter search was inspired by a previous neural network [63]. The neural network parameters were initialized *num_calibrations* (default: 100) times for each fold with weights suggested by hyperopt based on previous runs for the fold. The prior for the learning rate is log uniformly distributed over the interval [*epsilon_min, epsilon_max*] (default: [5e-4, 5e-2]). The prior for the convolutional filter weights is normally distributed with mean 0 and standard deviation *sigma_motifs*$^2$, where *sigma_motifs* is sampled from a log uniform distribution over the interval [*sigma_motifs_min, sigma_motifs_max*] (default: [1e-7, 1e-3]). Similarly, the prior for the fully connected layer weights is normally distributed with mean 0 and standard deviation *sigma_net*$^2$, where *sigma_net* is sampled from a log uniform distribution over the interval [*sigma_net_min, sigma_net_max*] (default: [1e-5, 1e-2]). The fully connected layer biases are initialized with the value 1e-5. The batch size was chosen by hyperopt from choices [64, 128, 256, 512]. Each network is trained for a number of epochs (default: 40). For each fold, the lowest validation set errors are calculated. The optimal network initialization is chosen as the network initialization with the lowest mean error for the validation set.

During the training stage, the network is initialized using the optimal network initialization (the learning rate, batch size, *sigma_motifs*, and *sigma_net*) for each fold. For each fold, the network is trained for the same amount of epochs as before on the inner training set, and network parameters for each fold are saved at the time point of the lowest validation set error. The network uses a mean squared error loss function, and the network parameters are optimized using the Adam optimizer. The network is trained with an early stopping strategy with patience 1. Prediction metrics are calculated for each fold separately using the prediction for the held-out "outer" test set.

### Alignment of motifs

To extract position weight matrices corresponding to the convolutional filters, activations are calculated for each model by convolving the convolutional filters across the training set sequences and passing the output through an ELU layer. Consequently, for each sequence, the subsequence of length *m* that generates the highest positive activation is isolated for each convolutional filter. Sequences that do not generate a positive activation score for a given convolutional filter are not included. For each convolutional filter, the scores are converted to position frequency matrices (PFMs) of size $4 \times m$ by summing the occurrences of each nucleotide at the corresponding sequence position. The PFMs are converted to position probability matrices (PPMs) by dividing each entry in the PFM by the sum of the nucleotide frequencies for the corresponding sequence position. The concatenated PPMs of the 10 models are then stored in a single MEME-formatted file. The motifs are aligned to MEME-formatted motif database files from CIS-BP [33] using the Tomtom tool from the MEME suite [64] using the argument -thresh 0.05. Sequence logos were visualized using the R package *ggseqlogo* [65].

### Leave-one-out analysis and influence scores

After training $r$ models, each model is reinitialized using the parameters from the optimal training point (the point at which the MSE on the validation set was the lowest). The prediction for motif $i$ in pool $j$ and sequence $k$ is denoted $b_{ijk}$. We then set all parameters related to motif $i$ to zero and calculate the perturbed score $c_{ijk}$ for each pool. The influence score for motif $i$ in pool $j$ is calculated as $n_{ij} = \frac{1}{N}\sum_{k=1}^{N}(b_{ijk} - c_{ijk})$ where $N$ is the number of sequences in the validation set. The influence scores from $d$ motifs from $K$ models are concatenated to get a matrix with $dK$ rows and $p$ columns. For each motif, the mean influence per cell type is calculated as an average from the corresponding pools. For visualization purposes, we also calculate a $z$-transformed version (based on cell types). Aligned motifs were assigned to a previously specified set of motif clusters [34]. To ensure reproducibility, motif clusters are excluded if they contain motifs from less than half of the $K$ models. Lastly, for most of the downstream analysis, motifs that did not align to known motifs are excluded. Influence scores for motifs that belong to the same motif family were summed to generate an "aggregate motif family influence" matrix with *num_motif_families* rows and $p$ columns. Similar to before, this matrix is then converted to include the mean aggregate motif family influence scores across cell types or cell type categories, and for visualization purposes, these scores are $z$-transformed.

### Assignment of putative transcription factors to motif clusters

For each motif cluster, the aggregate score in each pool was calculated as the sum of the motif influence scores in that pool for the motifs in that motif cluster. Each motif cluster has a list of potentially associated transcription factors, since these share similar binding motifs as determined by a previous motif clustering [34]. We assume that transcription factors with visually similar binding preferences are more likely to compete for binding when co-expressed, and as such, we aim to identify the degree to which transcription factors are contributing to the influence scores for a given motif cluster. We calculate the Spearman correlation between the aggregate motif cluster scores (across pools) and the expression of each of the potentially associated transcription factors (across pools). Transcription factor annotations for the top 3 associations were added using the R package *ggrepel*.

### Sequence "motif space" analysis

For the kidney and Tabula Muris experiments, promoter analysis was carried out by iterating over all sequences using the 10 models: the calculated scores were the outputs of the convolutional layer after ELU and max-pool operations for each model. The outputs were concatenated in the motif dimension to create an array of size (n_genes $\times$ 6000). "Motif space" UMAPs were calculated using this matrix using the R package *umap* [66].

### Proliferation and proximal tubule marker analysis

We used the following genes as markers of cell proliferation: MKI67, PLK1, E2F1, FOXM1, MCM2, MCM7, BUB1, CCNE1, CCND1, CCNB1, and TOP2A [67]. The

proliferation score of each pool was calculated as the average expression of this set of genes in that pool. Similarly, we used DLL1, ACAT1, PKD1, NOTCH2, AQP11, and HEYL for the proximal tubule signature genes.

### Gene Ontology analysis

The GO terms for mouse were downloaded from Ensembl Biomart. We used all categories with fewer than 50 genes for the analysis. For each GO term, the correlations between the expression of the GO term genes (across pools) and the aggregate motif influences of each motif family (across pools) were calculated. The top 500 GO terms in terms of their maximum correlation were hierarchically clustered based on Euclidean distance, after which the tree was cut with $k = 3$ to form the clusters.

### Analysis of co-occurring motif families

To investigate co-occurring motif families in the Tabula Muris dataset, the outputs of the convolutional layers after the ELU and max-pooling operations were first aggregated per motif family. Then, sets of genes with motif occurrence were selected for each motif family by filtering for genes with a score above 75% of the maximum gene score for that motif family. The upset plot was plotted using the R package UpsetR [68].

### Pseudotime analysis

The developing nephron compartment of the fetal kidney dataset was a subset of cells with a developmental relationship to cap mesenchyme (excluding ureteric bud). A two-dimensional UMAP embedding was calculated in scanpy [27] and was clustered using $k$-means clustering ($k = 10$). Pseudotime trajectories were computed with Slingshot [69] using the cluster corresponding to cap mesenchyme as a starting cluster. Proximal tubule pseudotime values were generated for cells along a continuous trajectory path from cap mesenchyme to the proximal tubule. The pseudotime values were averaged across the same cells that generated the pools in the combined human kidney dataset. In some cases, not all the cells in the pool had corresponding pseudotime values, as some cells fell outside of the $k$-means clusters along the principal curve corresponding to the proximal tubule pseudotime trajectory. The pseudotime values were only calculated for pools that had at least 100 cells (out of 120 cells per pool) with pseudotime values. The pooled (averaged) pseudotime values were subsequently normalized to range [0,1].

### Benchmark against FIMO motif regression and random forest

To benchmark scover to a linear regression ran on FIMO motif hits, we first ran FIMO using the database of all human motifs from CIS-BP for the human kidney and human brain datasets and using the database of mouse motifs from CIS-BP for the Tabula Muris dataset. The input sequences for FIMO were exactly the same as the inputs to the neural network, but in a fasta format. For each case, we used the FIMO default options with the command *fimo -o output_directory database.meme input_sequences.fa*. This generated a directory *output_directory* with (among other files) a table of match scores for each database motif in each sequence. We loaded the match scores into a large array *X_train* of size *num_sequences* by *num_motifs*, with a default score of 0 in case no motif match was found. We split the datasets up into the exact same train and test sets as the ones

used by scover and used python package scikit-learn to implement a linear regression on this data using the command *reg = LinearRegression().fit(X_train, Y_train)* where *Y_train* is the same data matrix that was predicted by scover as the target. We predicted the test set using the command *Y_pred = reg.predict(X_test)* and calculated the Pearson correlation between *Y_pred* and *Y_test*. We repeated this for each of the 10-folds of the data and used this to compare to the predictions made by scover. The workflow was the same for the random forest models, but instead of a linear regression model, the random forest was implemented using scikit-learn using command *reg = RandomForestRegressor(n_estimators = 20).fit(X_train, Y_train)*. This instantiated and trained a random forest regression with 20 estimators.

### Isolating genomic sequences

To prepare the sequences for the human kidney data, we isolated promoter sequences from the hg38 genome. We first used R commands "hg38 <- ensembldb::getGenomeTwoBitFile(EnsDb.Hsapiens.v86)" and "genes <- ensembldb::genes(EnsDb.Hsapiens.v86)" to extract the genome 2bit file and the list of genes and their genomic coordinates. We filtered the genes to only include those that were included in the single-cell RNA-seq dataset. We then used the functions "promoters <- GenomicRanges::promoters(genes, upstream = 500, downstream = 500)" and "sequences <- Biostrings::getSeq(hg38, promoters)" to extract 1000-bp promoter sequences around (500 bp upstream and 500 bp downstream) the transcription start sites (TSS). The TSS is defined in "GenomicRanges" to be the start index of the gene (taking into account gene orientation). We further removed any gene/sequence pairs that contained N nucleotides. For the Tabula Muris data, the above commands were the same, using "EnsDb.Mmusculus.v79" instead to extract the genome and genes associated with the mm10 genome. For the human brain dataset, we extracted the sequences for the genomic regions of the accessible sequences from "EnsDb.Hsapiens.v86." For each sequence, we calculated the middle of the sequence and extracted the 240-bp sequence that was 120 bp upstream and 120 bp downstream of the peak of the accessible site.

The input to scover (X_data) is the one-hot encoded sequence (of dimension sequence_length by 4 (for A, C, G, and T)). The to-be-predicted data (Y_data) is the pooled gene expression values associated with the promoter (for the expression model) or the pooled accessibility values associated with the accessible site (for the accessibility model).

### Data processing

#### Human kidney dataset

The fetal and adult kidney datasets were concatenated in the cell dimension to create a matrix of 33,660 features by 67,465 cells. A PCA was run and the first PC was ignored for further analysis as this correlated heavily with the total amount of RNA counts in the cell. The pooling strategy is described above. After pooling the datasets with $q = 120$ cells per pool, the dataset was filtered by excluding genes that were expressed in < 5% of pools. Another filter was applied: genes that were expressed in < 16% of pools were only retained if the mean non-zero log-transformed expression was higher than log10(3). This resulted in a dataset of 18,150 genes by 1000 pools. Running a full hyperparameter

sweep with cross-validation using $d = 600$ and $m = 12$ took a few hours when running across 8 NVIDIA Tesla V100 GPUs.

### Tabula Muris dataset

Twenty Tabula Muris FACS-sorted Smart-Seq2 datasets were concatenated in the cell dimension. A total of 297 erythrocytes were excluded from further analysis given their low counts, resulting in a matrix of 23,433 genes by 44,652 cells. The pooling strategy is described above. After pooling the dataset with $q = 80$ cells per pool, the data was filtered by removing genes that were expressed in $< 3\%$ of pools. Another filter was applied: genes that were expressed in $< 8\%$ of pools were only retained if the mean non-zero log-transformed expression was higher than log10(4). The final pooled dataset had a size of 17,270 genes by 1000 pools. When training the model, we used $d = 600$ and $m = 12$.

### Human brain dataset

The multimodal scRNA+ATAC-seq human brain dataset, containing 8981 cells and 467,315 accessible sites, was pooled as described above. After pooling the dataset with $q = 100$ cells per pool, the data was filtered by removing genes that were expressed in $< 6\%$ of pools. Only the loci that were outside the [$-8$ kb, 2 kb] region relative to annotated TSSs were considered. Another filter was applied: ATAC-seq regions that were detected in $< 15\%$ of pools were only retained if the mean non-zero log-transformed peak counts were higher than log10(4). The final pooled dataset had a size of 292,338 regions by 1000 pools. When training the model we used $d = 600$ and $m = 12$.

## Supplementary Information

---

**Additional file 1: Table S1.** Comparison of deep learning approaches for regulatory sequence signal prediction. **Table S2.** Assignment of cell types to categories for the human kidney dataset. **Table S3.** Assignment of cell types to categories for the Tabula Muris dataset.

**Additional file 2: Figure S1.** Statistics of the different datasets after pooling. (*a*) Largest cell type fraction per pool after pooling for the different datasets. (*b*) Number of cell types per pool for the different pooled datasets. (*c*) Total times a cell is included in pools for cells in the different datasets. (*d*) Fraction of non-uniquely assigned cells in a pool for the different pooled datasets. **Figure S2.** Correlations, number of cell types, sparsity (fraction of genes that are zero) and gene-wise coefficient of variation for influence scores (LOO gene CV) for different pool sizes for the datasets considered in this manuscript. The gene-wise coefficient for variation for influence scores was calculated only using reproducibly found motif families. **Figure S3.** (*a*) Example motif alignments for the human kidney dataset. (*b*) Random selection of 20 motifs with high impact scores that do not significantly align to the CIS-BP database. **Figure S4.** Pearson $R^2$ for the different experimental setups. Pearson $R^2$ for the held-out tests are shown for the models trained on the (*a*) human kidney, (*b*) Tabula muris, and (*c*) human brain datasets respectively. Some of the experiments on the x-axis reflect perturbation experiments, where either all motifs that aligned back to the database were left out (no non-aligned motifs), the order of the nucleotides in each sequence was permuted (scrambled sequences), or the order of each prediction vector was permuted (permuted cell pools). **Figure S5.** Projection of the pools onto the space represented by the first two principal components of the influence score matrix for the human kidney dataset. Color shows the summed influence scores for different motif clusters. **Figure S6.** Pairwise Pearson correlations between aggregate motif influence scores across pools between each of the reproducibly found motif clusters. **Figure S7.** (*a*) Example motif alignments for the Tabula Muris dataset. (*b*) Examples of randomly selected motifs with high influence scores that did not align to CIS-BP. Motifs 2_183 and 2_495 are examples of E-box-like motifs. **Figure S8.** (*a*) Aggregate motif weights of motif clusters averaged in cell types for the Tabula Muris dataset. (*b*) Z-transformed aggregate motif weights of motif clusters averaged in cell types for the Tabula Muris dataset. **Figure S9.** Co-occurring motif families in mouse promoters.

**Additional file 3.** Review history.

---

### Acknowledgements
We would like to thank Song Chen and Yingxi Lin for the helpful discussions about the SNARE-seq data and Irina Abnizova, Ilias Georgakopolous-Soares, Leopold Parts, and Nikos Patikas for the comments and suggestions on the manuscript and the package. We are also grateful to Simon Murray for testing the repository and to Jimmy Lee for sharing the code.

**Peer review information**

**Review history**

The review history is available as Additional file 3.

**Authors' contributions**

The project was conceived by JH, NKL, and MH. JH, NKL, and SR wrote the code. JH and MH analyzed the data. BJS and MRC assisted with the analysis of the kidney data. VC, MRC, and MH supervised the research. JH and MH wrote the manuscript with input from the other authors.

**Funding**

JH was supported by a grant for "Search tools for scRNA-seq data" (2018-183503) from the Chan Zuckerberg Initiative. MH, JH, and NKL were funded by a core grant from the Wellcome Trust, and JH received a PhD studentship from the Wellcome Trust. This research was funded in whole, or in part, by the Wellcome Trust [Grant number 206194 and Grant number 220540/Z/20/A]*. For the purpose of Open Access, the author has applied a CC BY public copyright license to any Author Accepted Manuscript version arising from this submission. MH, NKL, SR, and VC would like to acknowledge the funding from a Newton Mobility Grant (NI160206) from the Royal Society and the Thai Office of the Higher Education Commission.

**Availability of data and materials**

Code availability: Scover is freely available under the MIT license at GitHub (https://github.com/jacobhepkema/scover) [70] and Zenodo (https://doi.org/10.5281/zenodo.8169375) [71]. The code that was used to generate the figures can be found on GitHub (https://github.com/jacobhepkema/scoverplots) [72] and Zenodo (https://doi.org/10.5281/zenodo.8169168) [73].

Data availability: The human kidney datasets were downloaded from https://www.kidneycellatlas.org/ [74]. We downloaded 20 Tabula Muris FACS-sorted Smart-Seq2 datasets from https://doi.org/10.6084/m9.figshare.5829687.v8 [75]. The processed human brain dataset was downloaded using links from https://github.com/GreenleafLab/brainchromatin [76]. Pooled datasets can be accessed at (https://doi.org/10.5281/zenodo.8060659) [77].

## Declarations

**Ethics approval and consent to participate**

Not applicable.

**Competing interests**

The authors declare no competing interests.

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
