## [**Additional file 3.** Review history. · Genome Biology]

Review History

First round of review

Reviewer 1

Are you able to assess all statistics in the manuscript, including the appropriateness of statistical tests used? Yes, and I have assessed the statistics in my report.

Comments to author:

The authors use novel machine learning approaches to identify motifs that associate with active genes and accessible regions quantified from single cell omics data. The justification for using machine learning is that it is popular, but it is not clear what we learned from this approach that was not already known. In their conclusions they suggest that they quantify the relative importance of motifs, but I believe this is an overstatement. The motifs they find at promoters are well-known promoter motifs (the SP/KLG GC-rich motif was the first characterized motif in promoter bashing experiments—see McKnight and Kingbury, *Science* 1982) and conventional de novo motif analysis of transcription start sites identified these exact motifs in a cell-type agnostic way 9 years ago (see Benner, et. al., *PLOS Genetics* 2013). The statement that common regulatory patterns were revealed only means that the same motifs were identified across different sc*-seq experiments. I would be more positive about the manuscript if the authors pointed to one concrete new biological finding or simplified method development that helps discovery.

The input used for scover needs more explanation.

The TSS-proximal motif analysis within Figure 2 is not specific to fetal and adult human kidney and it is unclear why single cell data is necessary to identify these motifs. I refer the authors to Figure 1A of this paper from 9 years ago: <https://doi.org/10.1371/journal.pgen.1003906>—this publication identifies the same motifs when performing conventional de novo motif analysis on all promoters.

Scover explains 15% of the variance in gene expression within these cells, but they do not use a set of control scRNA-seq and scATAC-seq data to see if this explanation of variance is unique to the cell types that the model was trained upon.

This analysis and manuscript suggests that their work lends credibility to the notion of higher order organization of motifs, but as far as I can tell the authors do not inform on any of the principles of such a notion.

The authors do not mention how they define TF families that can "compete to bind similar motifs" when comparing relative expression levels of paralogous sequence-specific factors. Their methods do not go into details, but if they are only looking at weight matrices that are characterized, then they are missing TFs with paralogous DNA binding domains. If the authors choose to organize families in such a way, then TFClass is a resource that the authors may find useful.

Reviewer 2

Are you able to assess all statistics in the manuscript, including the appropriateness of statistical tests used? Yes, and I have assessed the statistics in my report.

Comments to author:

Summary:

This paper tries to identify motifs along with their cell-type importance from DNA sequences coming from single-cell assays measuring transcriptomic data such as scRNA-seq and chromatin accessibility such as scATAC-seq. They address the issue of sparsity in single-cell data by combining single cells into groups based on their similarity and learn PWM-style motifs using a CNN-based deep learning approach. For scRNA-seq, the gene expression is used as the supervision in the deep learning model.

This is an interesting paper and potentially very useful approach, the insights from their modeling will be useful to other researchers working on this topic. However, there are some issues that will need to be addressed before the work is ready for publication.

Major:

1. The results of the pooling will be influenced by the types of DNA sequences used - i.e. promoters / enhancers or all genomic regions with some expression / accessibility. The use of all genomic regions as input to the clustering, might not produce as distinct clusters as is needed by the subsequent steps of this approach. Is the assumption that all locations of interest are known for the cell-types being studied? Like from a DNase-1 hypersensitivity atlas? Is this a drawback of this approach, that it can only be applied to such regions?
2. To assign motifs to TFs, can the authors consider an approach like Algorithm-1 in this paper BindVAE <https://genomebiology.biomedcentral.com/articles/10.1186/s13059-022-02723-w> Can you do inference on SELEX probes (in vitro TF binding data) and look at the activation of the CNN's filters? The highest activation for a SELEX TF's probes can indicate that filter/PWM has learned those patterns.
3. To get the importance of each motif, an "influence score" is computed as a post-processing step. Why can't the pool-specific weight matrix W learned by the model be used?
4. What about matching the motif to all sequences in a pool to calculate the influence score? Can the authors plot that shows how much the sum(motif matches) varies across the pools?
5. The % of convolutional filters matching to motifs seems to be very small - 8% (and in that range for other datasets). Why is this the case? There is some literature that suggests that multiple filters together capture coherent patterns, i.e. the information from a motif's PWM is spread across multiple conv-net filters. Have the authors considered looking into this?
6. Re: match % of convolutional filters, as a comparison, the BindVAE paper (reference above) seems to find 60% matches (60 or so from 100) for the learned latent topics to motifs. Does this

suggest that k-mer distributions as proposed by the BindVAE paper are better representations than PWM matrix like filters from CNNs?

7. Have the authors tried different activation functions? For instance, using exponential instead of ReLU right after the filters, it might result in more motif-like filters being learned.

8. Why are so many convolutional filters redundant? Is there a way to encode for GC-rich motifs in the model that will not cause these regions from dominating what's learned by the model?

9. Do the authors find any patterns that represent experimental artifacts like background genomic patterns or enzyme cleavage bias. For instance, with ATAC-seq data, Tn5-transposase cleavage bias has been found in processed reads. Or are these artifacts removed due to their consideration of only known promoter regions?

10. The authors find 11 motif families in the human kidney dataset - are there particular TF families (say homeodomains or bZIPs) that are large and easy to find across the datasets tried? Likewise, are there others that are more difficult / rare? If so, can they mention this in their results?

11. Can the authors cite literature for the assumption that TF expression relates to its activity. Can an approach that simply takes known expressed TFs and matches their motifs using algorithms like FIMO do just as well as the proposed CNN approach? What are the merits of using the proposed model?

12. There is no comparison to motif-matching methods like FIMO or other motif discovery algorithms like HOMER, MEME or recent deep learning motif finding methods like BindVAE.

13. Have they considered showing the results of a simple approach like motif-matching using FIMO on the DNA sequences and regressing the motifs against the observed expression? The weights of this regression problem can be considered as motif influence scores.

14. It seems that there are cases where a TF has a high influence score, but none of the TFs in the corresponding family is expressed. Can the authors explain in detail why this might happen? Can they show a plot of Expression (y-axis) and Influence-score (x-axis)?

15. Is the influence score matrix processed by normalizing? Can they show a PCA of individual cells with just the input expression values, i.e. PCA of input data to get an understanding of how much improvement in clustering is obtained by the model?

16. Citations: Some relevant papers to cite

Fang, R. et al. Comprehensive analysis of single cell ATAC-seq data with SnapATAC. Nature Communications 12, 1337 (2021)

Kshirsagar M, et al. BindVAE: Dirichlet variational autoencoders for de novo motif discovery from accessible chromatin. Genome biology (2022)

Haghverdi, L., Lun, A.T.L., Morgan, M.D. & Marioni, J.C. Batch effects in single-cell RNA-sequencing data are corrected by matching mutual nearest neighbors. Nat Biotechnol

17. Have the authors considered incorporating longer-range dependence into the model? For instance, using dilated convolutions? Would this be helpful?

Minor:

In choosing the number of pools, is the hope that the number of pools == number of cell types? What about overlapping pools, where a cell is present in several pools? And what about cells from a single cell type being scattered across several different homogenous pools? Or worse, heterogeneous pools?

Page-7, Line-35: summing over all motifs in a cluster will usually be a large positive value. Also, summing over all pools will have the same issue. Are there any cases where this value is a small number? Does averaging make more sense?

What is the distance measure used to compute similarity between the cells by the Pooling/sketching algorithm? Was UMAP dimensionality reduction used to get the embeddings for the single cells?

How was the number of 1000 seed cells for the human kidney dataset decided or 100 cells for the P0 dataset? Was it based on the size of the dataset (number of cells sequenced), or depth/ quality of the reads?

Is the sampling of cells going to be sensitive to experimental variation - for ex: cells from the same sequencing depth will cluster together. How do they address other such factors?

It would be good to explain the biological motivation behind the bias term - what other factors that are not TFs influence the outcome?

On the human kidney dataset, it is not clear what the plots in Fig 3(a) are depicting. It seems like most promoter regions that have the motif being shown are randomly distributed in the plot?

The observed values "t_obs" - is it the average of values from a pool?

Why did they use 10 CNN models - what variability does each represent?

"...biggest difference in influence correlated...". Please mention the difference in influence of?

We would like to thank the reviewers for their constructive comments and the editor for allowing us to resubmit our manuscript. We are also grateful for the multiple extensions of the resubmission deadline that we were granted. We have addressed all of the issues raised by the reviewers, and we believe that this has strengthened the manuscript significantly. In the updated manuscript, all changes to the text are marked in red. The manuscript has undergone substantial revisions and, among other things, two major updates to the previous version of the manuscript are the replacement of the multimodal dataset used for training the ATAC-seq model, and the switch from rectified linear units to exponential linear units across all models. Please find point-by-point responses below:

Reviewer #1: The authors use novel machine learning approaches to identify motifs that associate with active genes and accessible regions quantified from single cell omics data. The justification for using machine learning is that it is popular, but it is not clear what we learned from this approach that was not already known. In their conclusions they suggest that they quantify the relative importance of motifs, but I believe this is an overstatement. The motifs they find at promoters are well-known promoter motifs (the SP/KLG GC-rich motif was the first characterized motif in promoter bashing experiments—see McKnight and Kingbury, Science 1982) and conventional de novo motif analysis of transcription start sites identified these exact motifs in a cell-type agnostic way 9 years ago (see Benner, et. al., PLOS Genetics 2013). The statement that common regulatory patterns were revealed only means that the same motifs were identified across different sc*-seq experiments. I would be more positive about the manuscript if the authors pointed to one concrete new biological finding or simplified method development that helps discovery.

While most of the motifs we identify are indeed enriched in promoters and can be identified in a cell-type-agnostic way (as the promoters stay the same), our model carries out a multiple regression using the motifs that are found across the different cells. As such, it can identify how important a given motif is for the prediction in a given cell type. Some motif importance scores will hardly have any variation across these cell types, whereas others vary substantially. As such, this gives an additional dimension of interpretation over just the motif enrichment analysis - motif enrichment alone as in the manuscripts cited by the reviewer cannot identify such cell type specific differences.

With regards to the reviewer's last point regarding novelty, we highlight two aspects:

1. Scover enables simultaneous de novo motif finding (which can also leverage existing motif annotation if available) and identification of the cell type specific weight. To the best of our knowledge, there is no other method available that can carry out these two tasks simultaneously. The ability to carry out these two tasks together simplifies the workflow compared to the alternative where motifs are first inferred and then subsequently used in a regression model to determine their contribution to gene expression.

2. To the best of our knowledge, the results reported in Fig 5b constitute a new biological finding. To recap, our global analysis of the TSS proximal regulatory landscape across 20 mouse tissues revealed three distinct groups of motif activities and functional categories. This association is indicative of coordinated regulation, not just across a single pathway, but across broad functional categories. Although other studies have reported findings that are conceptually similar (see p6), there are details in our manuscript that are not present in the other studies.

The input used for scover needs more explanation.

We apologize for the lack of clarity. We have added clarification in the methods section.

Pooling

Scover was trained as a regression model to predict the log-transformed pooled gene expression values (a vector of size p) from the associated promoter sequence of the gene. To reduce the sparsity, pools are created by grouping k -nearest neighbors of an initial selection of cells ('seed cells') calculated using a shared PCA embedding into p equally sized groups of q cells (Fig 1a). The pooled expression values were calculated as follows: for a given single-cell dataset, a PCA representation was calculated based on the log-transformed counts of the data. For all datasets, the first PC was not used for subsequent steps since it correlated highly with the number of counts in the pools. As the RNA datasets consist of concatenated datasets of different batches, batch-balanced k -nearest neighbor algorithm BBKNN was run with default parameters, which was used subsequently to calculate a batch-balanced UMAP dimensionality reduction. Using geometric sketching, a subset of p cells (the 'seed cells') was selected that spans the UMAP representation of the dataset in a uniform way (1). Then, k -nearest neighbors of these initial 'seed cells' were calculated again, but now k was set to be the pool size. For each of the seed cells, the pooled counts were calculated as the sum of the raw counts for the seed cell and its k -nearest neighbors, followed by a $\log(1+\text{counts})$ transformation. Before training, the datasets were further filtered in a dataset-specific way as described in section 'Data processing' below. A second task we applied Scover to was to predict the log-transformed pooled chromatin accessibility values (again, a vector of size p) from the associated accessible region sequences. For this, we used multimodal scRNA+ATAC-seq data, so that we can compare learned motif influence scores to the transcription factor expression. The approach was exactly the same as for the scRNA-seq data, but here we re-use the exact same neighbors to pool both modalities, and no batch correction was applied. In this case, the PCA representation, k -nearest neighbors, and the UMAP representations were calculated using the transcriptomic representation of the data. Before training, the dataset was further filtered in a dataset-specific way as described in section 'Data processing' below.

Isolating genomic sequences

To prepare the sequences for the human kidney data, we isolated promoter sequences from the hg38 genome. We first used R commands 'hg38 <- ensemblDb::getGenomeTwoBitFile(EnsDb.Hsapiens.v86)' and 'genes <-

ensemblDb::genes(EnsDb.Hsapiens.v86)' to extract the genome 2bit file and the list of genes and their genomic coordinates. We filtered the genes to only include those that were included in the single-cell RNA-seq dataset. We then used the functions 'promoters <- GenomicRanges::promoters(genes, upstream=500, downstream=500)', and 'sequences <- Biostrings::getSeq(hg38, promoters)' to extract 1000 bp promoter sequences around (500 bp upstream and 500 bp downstream) the transcription start sites (TSS). The TSS is defined in 'GenomicRanges' to be the start index of the gene (taking into account gene orientation). We further removed any gene/sequence pairs that contained N nucleotides. For the tabula muris data, the above commands were the same, using 'EnsDb.Mmusculus.v79' instead to extract the genome and genes associated with the mm10 genome. For the human brain dataset, we extracted the sequences for the genomic regions of the accessible sequences from 'EnsDb.Hsapiens.v86'. For each sequence, we calculated the middle of the sequence and extracted the 240 bp sequence that was 120 bp upstream and 120 bp downstream of the accessible site.

The input to scover (X_data) is the one-hot encoded sequence (of dimension $sequence_length$ by 4 (for A, C, G, and T)). The to-be-predicted data (Y_data) is the pooled gene expression values associated with the promoter (for the expression model), or the pooled accessibility values associated with the accessible site (for the accessibility model).

Data processing

Human kidney dataset

The fetal and adult kidney datasets were concatenated in the cell dimension to create a matrix of 33,660 features by 67,465 cells. A PCA was run and the first PC was ignored for further analysis as this correlated heavily with the total amount of RNA counts in the cell. The pooling strategy is described above. After pooling the datasets with $q = 120$ cells per pool, the dataset was filtered by excluding genes that were expressed in $< 5\%$ of pools. Another filter was applied: genes that were expressed in $< 16\%$ of pools were only retained if the mean non-zero log-transformed expression was higher than $\log_{10}(3)$. This resulted in a dataset of 18,150 genes by 1,000 pools. Running a full hyperparameter sweep with cross-validation using $d = 600$ and $m = 12$ took a few hours when run across 8 NVIDIA Tesla V100 GPUs.

Tabula Muris dataset

20 Tabula Muris FACS-sorted Smart-Seq2 datasets were concatenated in the cell dimension. 297 erythrocytes were excluded from further analysis given their low counts, resulting in a matrix of 23,433 genes by 44,652 cells. The pooling strategy is described above. After pooling the dataset with $q = 80$ cells per pool, the data was filtered by removing genes that were expressed in $< 3\%$ of pools. Another filter was applied: genes that were expressed in $< 8\%$ of pools were only retained if the mean non-zero log-transformed expression was higher than $\log_{10}(4)$. The final pooled dataset had a size of 17,270 genes by 1,000 pools. When training the model we used $d = 600$ and $m = 12$.

Human brain dataset

The multimodal scRNA+ATAC-seq human brain dataset, containing 8,981 cells and 467,315 accessible sites, was pooled as described above. After pooling the dataset with $q = 100$ cells per pool, the data was filtered by removing genes that were expressed in $< 6\%$ of pools. Only the loci that were outside the $[-8 \text{ kb}, 2\text{kb}]$ region relative to annotated TSSs were considered. Another filter was applied: ATAC-seq regions that were detected in $< 15\%$ of pools were only retained if the mean non-zero log-transformed peak counts were higher than $\log_{10}(4)$. The final pooled dataset had a size of 292338 regions by 1000 pools. When training the model we used $d = 600$ and $m = 12$.

The TSS-proximal motif analysis within Figure 2 is not specific to fetal and adult human kidney and it is unclear why single cell data is necessary to identify these motifs. I refer the authors to Figure 1A of this paper from 9 years ago:

https://secure-web.cisco.com/1XphLW7gAbyTgdYnxctauzW3ZbFkF64a-bF5b6gB0pdnv19Nnhof-FaJRQpPxzoBDZniWcp5SAfwdDKzfJF7Yfqg3yB0qUuawLk-4slfecuixFkdjHZj2JAAprTDE2nJ4tW0A1E97FcX_T1IIFkcDWIn7AEIX0vE94Q8GeFW7TprAx9EhCgSENI9EnyuWyPX9Rxx_KKUJGkG_lh1nsPY_pDT0jWUdOSfNozCRUP0LC3ZB9cvF5tjnOVqW9iVCe6w51_W3mfb5VF8u5ebHohVs2HMLiknjJ86jOeCY2OrN-bqTJCSP2fx9T74Wtlivetlr/https%3A%2F%2Fdoi.org%2F10.1371%2Fjournal.pgen.1003906—this publication identifies the same motifs when performing conventional de novo motif analysis on all promoters.

The reviewer makes a good point regarding the limited benefits of using single cell data for studying the kidney as it is a very well characterized organ. We agree that for the purpose of identifying key motifs there is no major advantage to using single cell data. However, we view this as a positive control and the result should instill confidence that motifs obtained from the analysis of other single cell data sets will be reliable. More importantly, our method goes one step beyond motif identification and it also determines the relative contribution in each cell type of each motif towards gene expression levels. With the bulk data, it is not clear what the cell type specific role of each motif is. Nevertheless, we have referenced the Benner et al paper on p5 and commented on the fact that they were able to uncover similar motifs using older technologies.

Scover explains 15% of the variance in gene expression within these cells, but they do not use a set of control scRNA-seq and scATAC-seq data to see if this explanation of variance is unique to the cell types that the model was trained upon.

The reviewer makes a good point regarding the cell-type-specific explanation of variance. We have addressed this issue by carrying out additional control experiments whereby we also permuted the order of the pools in the prediction vector. We expect that a lower fraction of variance explained reflects that the model learned cell type-specific signal; conversely, if the fraction of variance explained does not go down, this implies that the order of the pools when correlating to predicted scores is unimportant. As expected, we observe a significant drop in the fraction of variance explained for the kidney and mouse datasets (**Fig R1**), suggesting that

grouping cells by their transcriptional/chromatin similarity is crucial to performance. However, for the SNARE-seq data, we only observed a small, but non-significant decrease in accuracy, suggesting that scover is unable to find a cell-type specific model for this dataset. Because of this failed control experiment, we have removed the SNARE-seq results from the new version of the manuscript. Instead, we ran scover on a different dataset, from developing human cerebral cortex (2). We refer to this dataset throughout as 'human brain' and we have entirely rewritten the last section of the results. We suspect that the reason for the poor performance of scover on the SNARE-seq dataset is down to quality of the data. The SNARE-seq dataset was one of the first joint scRNA/ATACseq datasets ever published and thus it is not surprising that a later dataset, which relied on a commercial 10X kit, achieves substantially better quality data.

Figure R1: Controls for scover models. Pearson R2 for the different experimental setups. Pearson R2 for the held-out tests are shown for the models trained on the (a) human kidney, (b) Tabula muris, (c) human brain, and (d, e) SNARE-seq data respectively. Some of the experiments on the x-axis reflect perturbation experiments, where either all motifs that aligned back to the database were left out (no non-aligned motifs), the order of the nucleotides in each sequence was permuted (scrambled sequences), or the order of each prediction vector was permuted (permuted cell pools). Please note that the SNARE-seq plots show the Pearson R rather than R^2 . For (a), (b) and (c), results are also shown for selected experiments using FIMO motif regression.

This analysis and manuscript suggests that their work lends credibility to the notion of higher order organization of motifs, but as far as I can tell the authors do not inform on any of the principles of such a notion.

The reviewer raises an excellent point and it is indeed one of our messages that there is a higher order organization of motifs. This concept is not a new one, and it has been raised in several publications. Interestingly, varying explanations, including chromatin structure and evolution, have been put forward to explain these principles. We have updated the discussion to add a paragraph (please note that numbers for references are different here compared to main text):

Our finding that motif influence scores are related to genes associated with specific pathways findings (Fig 5b) is indicative of higher order organizational principles determining the motif arrangement in promoter regions. This is consistent with previous studies which have demonstrated that regulatory motifs occurrences at promoters are associated with specific GO terms (3,4). Similarly, cross-species comparisons have shown that complex behavioral phenotypes are associated with specific transcription factor motifs (5) (6). The underlying mechanism is most likely the 3D organization of the chromatin (7) although it is unclear to what extent it is directly causal.

The authors do not mention how they define TF families that can "compete to bind similar motifs" when comparing relative expression levels of paralogous sequence-specific factors. Their methods do not go into details, but if they are only looking at weight matrices that are characterized, then they are missing TFs with paralogous DNA binding domains. If the authors choose to organize families in such a way, then TFClass is a resource that the authors may find useful.

We apologize for the lack of clarity here, and we have added more information in the methods subsection 'assignment of putative transcription factors to motif clusters'.

Assignment of putative transcription factors to motif clusters

Each motif cluster has a list of potentially associated transcription factors that recognise the same motif. We defined TFs that compete to bind similar motifs to be TFs in the same motif cluster as defined by a previous study (8), https://www.vierstra.org/resources/motif_clustering),

where different TFs are clustered based on their visual motif similarity. We have not quantified the degree to which TF families can compete to bind similar motifs, but we assume that transcription factors with visually similar binding preferences are more likely to compete for binding when co-expressed. With this analysis, we aim to identify the degree to which transcription factors are contributing to the influence scores for a given motif cluster. We calculate the spearman correlation between the aggregate motif cluster scores (across pools) and the expression of each of the potentially associated transcription factors (across pools). We visualize the associations between motif clusters and TFs in a scatterplot, where the transcription factor annotations for the top 3 associations were added using R package *ggrepel*. We appreciate the reviewer's suggestion to use TFClass. Our understanding is that TFClass allows the identification of TFs that have no known binding motif, but similar binding domains. This could potentially help find additional TF candidates, e.g. based on correlations of the expression levels with influence scores. We have added a brief mention of this possibility in the discussion.

Reviewer #2: Summary:

This paper tries to identify motifs along with their cell-type importance from DNA sequences coming from single-cell assays measuring transcriptomic data such as scRNA-seq and chromatin accessibility such as scATAC-seq. They address the issue of sparsity in single-cell data by combining single cells into groups based on their similarity and learn PWM-style motifs using a CNN-based deep learning approach. For scRNA-seq, the gene expression is used as the supervision in the deep learning model.

This is an interesting paper and potentially very useful approach, the insights from their modeling will be useful to other researchers working on this topic. However, there are some issues that will need to be addressed before the work is ready for publication.

Major:

1. The results of the pooling will be influenced by the types of DNA sequences used - i.e. promoters / enhancers or all genomic regions with some expression / accessibility. The use of all genomic regions as input to the clustering, might not produce as distinct clusters as is needed by the subsequent steps of this approach. Is the assumption that all locations of interest are known for the cell-types being studied? Like from a DNase-1 hypersensitivity atlas? Is this a drawback of this approach, that it can only be applied to such regions?

We apologize for the lack of clarity and the confusion created here. Our method does not assume that all locations of a specific class of interest are known. Instead, it can take as input any set of DNA sequences that are associated with a vector of real values (in the paper we have used expression levels and ATACseq reads). Clearly, if only a subset of regions (e.g. promoters or candidate enhancers) are included, then this will limit the conclusions that one is able to draw, but it is not an intrinsic limitation of the method. In the case of predicting chromatin accessibility data, it is easy to adapt it to include more genomic regions by adding these regions

and their corresponding signal (zero). We have clarified this point in the first paragraph of the discussion.

2. To assign motifs to TFs, can the authors consider an approach like Algorithm-1 in this paper BindVAE

https://secure-web.cisco.com/1s6JPY7T8OCkwmd2TwiGfMwlrI8YpPDP4z93J_BMAyUEtBqKDuAHZ0LDqW49piLDCLDeRTYJdc-28VsSP6DIYq3otuCBaml358sAjo4ZSWHlye0TgAff_z91bFyOsZOgDY3y2Mr5JJN9W7hOcn3pqpGHhL8TkQITWxPBT581rRXEvaBvW6ASWeL4Zaw-m3dliB8dKLDLVb0Z6-fTxmbVb5zOKbHfpEGHqU1m_OmJuJsrhWCnEzafQcofq0vGhp4hewIYIAKu5uB0P0qTcWjsbdkcuJqt38RKqFgqxI9_31AqUPUFXrrA3TszuAr3wSLz/https%3A%2F%2Fgenomebiology.biomedcentral.com%2Farticles%2F10.1186%2Fs13059-022-02723-w Can you do inference on SELEX probes (in vitro TF binding data) and look at the activation of the CNN's filters? The highest activation for a SELEX TF's probes can indicate that filter/PWM has learned those patterns.

We thank the reviewer for this suggestion and we have analyzed the same SELEX data as in the BindVAE manuscript.

We used the Tabula Muris data to obtain convolutional filters for scover. For each convolutional filter we evaluated the outputs of the convolutional scan after the exponential linear unit and max-pooling when convolving over all of the SELEX probes. Following a similar method to Algorithm 1 from the BindVAE paper, we then performed a one-vs-all Mann-Whitney U test to ask if the score is significantly higher for a specific convolutional filter, followed by Benjamini-Hochberg multiple testing correction.

After applying the algorithm, the median number of TFs significantly associated ($p < 0.05$) with each filter is 49, suggesting that the mapping is not specific. Indeed, when we inspect a few random significant associations between convolutional filters and TFs as determined by this algorithm, we find that the sequence logos are not very similar (**Fig R2**). For a given convolutional filter, we find that there is a mix between associated TFs that make sense and TFs that do not. As such, we conclude that this algorithm does not help in narrowing down the potentially associated TFs.

Figure R2: Example significant associations between motifs and SELEX TFs. TF motifs were taken from JASPAR (9).

We conjecture that there are a few explanations for the poor performance of this approach as compared to BindVAE. First, the scover model was trained on in vivo data and the procedure uses in vitro data. Hence, we suspect that when trained on the more complex in vivo data the model performs worse with the cleaner in vitro data. Second, BindVAE was trained on ATACseq data from a cell line, and this implies that the sequences used are much more diverse than the promoters that were used by scover. Third, the main goal of scover is not to find an optimal representation of motifs, but to predict gene expression (or other signal). Clearly, a good representation is beneficial for this task, but it is not what has been optimized for.

3. To get the importance of each motif, an "influence score" is computed as a post-processing step. Why can't the pool-specific weight matrix W learned by the model be used?

The reviewer raises an important question, and this used to be our approach. However, since the values in the convolutional filters can be of different distances to 0, and the subsequent nonlinearity and max-pool operations do not reduce this scale (on the positive end of the spectrum), the outcome of the matrix multiplication of the convolution output with weight matrix W depends on the scale of the convolutional filter. This is why individually perturbing the motifs, by setting either the convolutional filters to 0 or the corresponding row in pool-specific weight matrix W to 0, and then looking at the difference in the output is a better way of assessing the influence of a convolutional filter on the prediction, since this takes the scale of the convolutional filter into account.

4. What about matching the motif to all sequences in a pool to calculate the influence score? Can the authors plot that shows how much the sum(motif matches) varies across the pools?

We appreciate the reviewer's suggestion, and we apologize for the confusion. Our algorithm finds the same motifs for all pools and it is only the weights that are different. Hence, there is no point in showing the sum(motif matches) across pools since the value will be the same. We have clarified this point in the text on p5.

5. The % of convolutional filters matching to motifs seems to be very small - 8% (and in that range for other datasets). Why is this the case? There is some literature that suggests that multiple filters together capture coherent patterns, i.e. the information from a motif's PWM is spread across multiple conv-net filters. Have the authors considered looking into this?

We thank the reviewer for bringing up an important issue. For the new version of the manuscript, in which we use exponential linear units instead of rectified linear units, the percentage of motifs that significantly ($p < 0.05$) aligned to a database motif is 13.3% for the human kidney model, 12.0% for the tabula muris model, and 24.5% for the Trevino et al human brain dataset. While this is an improvement, these percentages are still low. We think that this is the case because there are multiple unfavorable local minima where the optimization procedure gets stuck. We assume that good initialisations for the convolutional filters matter for their capacity to converge to a meaningful motif, and by having many convolutional filters, we hope to be able to increase the chances of finding a large amount of the relevant motifs.

With regards to the second question regarding multiple patterns forming a motif, this is indeed something that we have considered. The main reason why we think that this is not the case for our model is because the default size of the convolutional filters was deliberately set to be greater than the vast majority of known motifs in human and mouse. Hence, with the exception of large motifs such as CTCF, we think it is very unlikely that there are motifs that cannot be found by a single filter using our current strategy. The reason for this design choice was the study by Koo and Eddy (10) which concluded that NN models provide more interpretable solutions when a localist representation is used. Although we have not tried to quantify this aspect, visual inspection of the identified motifs did not suggest that the model identified distributed motifs (other than GCs). Indeed, having a single convolutional layer plus one linear layer only allows for a linear combination of the features from the convolutional layer - the single linear layer cannot implement AND logic. Since an architecture that combines multiple convolutional filters will require more layers and parameters, we decided against this approach at an early stage of the project.

6. Re: match % of convolutional filters, as a comparison, the BindVAE paper (reference above) seems to find 60% matches (60 or so from 100) for the learned latent topics to motifs. Does this suggest that k-mer distributions as proposed by the BindVAE paper are better representations than PWM matrix like filters from CNNs?

The reviewer raises an important issue regarding representation of TFBSs. We are aware of the debate regarding the use of kmers vs PWMs. There are advantages over PWMs as inferred by single-layer CNNs like ours, for instance that they capture within-motif nonlinearities (11). However, the number of convolutional filters that match motifs in the database is an imperfect measure of how good the PWM-like representation is: it merely reflects how many of the convolutional filters, including redundant ones, managed to converge to a motif that aligns to the database motifs. Next to this, the low % of convolutional filters aligning does not per se mean a failure to capture a variety of motifs.

Empirically, we have found that using many convolutional filters helps find motif families that we would otherwise not find. The drawback of this strategy is of course that it is computationally more costly and we agree with the reviewer that finding a better representation is an important future direction.

To address the specific question, we again want to emphasize that our optimization strives to maximize the % of variance explained rather than the number of matching filters. During development, we explored different numbers of filters and we found that when a smaller number was used then the fraction matching known motifs was higher, but with lower % of variance explained. Importantly, when a smaller number of filters was used, the number of motif clusters we were able to reproducibly detect (after alignment) was lower as well.

7. Have the authors tried different activation functions? For instance, using exponential instead of ReLU right after the filters, it might result in more motif-like filters being learned.

We thank the reviewer for this helpful suggestion. We have implemented this activation function and as a result all 6,000 filters generated a motif-like output after training for all three datasets, likely because there are fewer vanishing gradients since they allow for negative output as well. Based on this result, we have changed our model so that the exponential LU is the default, and we have re-created all of the results in figure 2-6 (and the relevant supplementary figures). Although the overall results are robust, several details throughout the manuscript have changed.

8. Why are so many convolutional filters redundant? Is there a way to encode for GC-rich motifs in the model that will not cause these regions from dominating what's learned by the model?

In the current version of the model, there is nothing preventing the model from learning multiple convolutional filters that correspond to the same transcription factor. This is why we perform most analyses on the motif family level, such that the influence scores of convolutional filters with similar patterns are analyzed together. However, our new analyses (see pt 10 below) indicate that the motifs found are not primarily driven by low level features, suggesting that the GC-rich motifs are of interest and we are not convinced that it would be beneficial to forcibly exclude them. We are aware of approaches for enforcing dissimilarity of learned features (12) (13), but this is outside the scope of the current work.

9. Do the authors find any patterns that represent experimental artifacts like background genomic patterns or enzyme cleavage bias. For instance, with ATAC-seq data, Tn5-transposase cleavage bias has been found in processed reads. Or are these artifacts removed due to their consideration of only known promoter regions?

The reviewer raises an important question and we agree that there could be Tn5-transposase cleavage bias present in the processed reads. Unfortunately, despite extensive searches, we were unable to find a PWM representation of the Tn5 motif. Since the best we could find was a graphical representation (Fig 2 from (14)) we resorted to visual inspection of randomly selected non-aligned motifs. Albeit fraught with shortcomings, manually inspecting the top 20 (in terms of their absolute motif influence scores) unaligned motifs for the human brain dataset suggested that some could represent partial Tn5 recognition motifs (**Fig R3**). We expect that the potential Tn5 motifs have a slight influence on the explained variance metric. However, since we restrict our downstream analysis on the aligned motifs only, we do not expect this to influence our main findings.

Fig. R3. Top 20 motifs in terms of the absolute motif influence scores for the human brain dataset. Motifs 0_452 and 6_337 contain similar patterns to the Tn5 cleavage bias motif.

10. The authors find 11 motif families in the human kidney dataset - are there particular TF families (say homeodomains or bZIPs) that are large and easy to find across the datasets tried? Likewise, are there others that are more difficult / rare? If so, can they mention this in their results?

We thank the reviewer for this suggestion and we have quantified the motifs by their complexity. We defined motif complexity by the information content (IC) and we calculated it from the position probability matrices using the function 'transform_matrix' from python package *logomaker*. We then correlated the IC with the summed influence scores for each motif (**Fig R4**), and the results show that there is no positive correlation between the two quantities. This result strongly suggests that our algorithm is not biased by low level properties.

Figure R4: Motif information content and influence scores. 2D histogram showing the summed motif information content (x-axis) versus the summed influence score for the motif (y-axis), for the models trained on the human kidney dataset (Pearson R = -0.25), Tabula Muris dataset (Pearson R = -0.24), and the human brain dataset (Pearson R = -0.12).

Furthermore, the following motif (sub)families that were consistently found with models trained using the expression datasets:

- ZFX:C2H2
- KLF/SP/2:C2H2
- Ebox/CACGTG/1:bHLH
- ETS/1:ETS
- ETS/2:ETS
- CREB/ATF/1:bZIP
- YY1:C2H2
- Ebox/CACCTG:bHLH
- Ebox/CAGCTG:bHLH
- GC-tract:C2H2
- E2F/2:E2F

These motif families are considered 'found' in a training run when at least half of the 10 models trained align any motif to the motifs in the motif subfamily.

The following motif (sub)families were only found in the model trained on tabula muris data:

- Ebox/CACGTG/2:bHLH
- NR/1:nuclearreceptor

Similarly, the following motif (sub)families were only found in the model trained on human kidney data:

- GLI:C2H2
- NRF1:CNC-bZIP
- KAISO:C2H2
- CTCF:C2H2

This can be the result of the fact that the promoter sequences in mouse and human are different in motif composition. However, it might also point towards the ease of finding particular motif families given the right data; for instance, GLI transcription factors are associated with the hedgehog signaling pathway, which is important in kidney development (15).

The following motif (sub)families were found in both expression and accessibility datasets:

- KLF/SP/2:C2H2
- CREB/ATF/1:bZIP
- GC-tract:C2H2
- Ebox/CACCTG:bHLH
- CTCF:C2H2
- Ebox/CAGCTG:bHLH
- NR/1:nuclearreceptor

There were many motifs (sub)families only found using the accessibility dataset, but this is not surprising since the human brain data included distal sequences that are much more diverse than promoters.

11. Can the authors cite literature for the assumption that TF expression relates to its activity. Can an approach that simply takes known expressed TFs and matches their motifs using algorithms like FIMO do just as well as the proposed CNN approach? What are the merits of using the proposed model?

Models of enhancers (see e.g. Spitz and Furlong, Nat Rev Gen, 2012) where the activity is proportional to TF concentration (which can be assumed to be proportional to the expression level) have been proposed, and there is experimental data to support the model for NF-KappaB (Giorgetti et al, Mol Cell, 2010). We also assume that a transcription factor needs to be expressed in order for it to be active, and using the correlation, we hope to capture how well the pattern of it being expressed across cell pools (regardless of the level of expression) matches how much influence the motif has in the cell pools.

One advantage of using our model over using FIMO is that (as discussed below point 13) learnable motifs followed by a nonlinearity allow us to adapt the convolutional filter weights to find the threshold for when a motif match is deemed significant. In other words, the network can

adapt the weights to determine how good a motif match should be to cross the threshold of the nonlinearity.

12. There is no comparison to motif-matching methods like FIMO or other motif discovery algorithms like HOMER, MEME or recent deep learning motif finding methods like BindVAE.

We appreciate the reviewer’s suggestion and we have carried out additional comparisons. However, we would like to emphasize that scover differs from the methods suggested as it not only finds enriched motifs, but it also determines their contribution to gene expression. Nevertheless, finding high quality motifs is a key requirement for the subsequent task of predicting motif influence scores.

For the Tabula Muris dataset, we compared against HOMER in the following way: using the pooled dataset, we selected the promoter sequences (the same as the input to our model) of the top 500 genes that are differentially expressed in the immune cell category (using the t-test method of scanpy’s ‘rank_genes_groups’ function). The sequences of all other promoters were used as background (n=20218). The choice of 500 DEGs is arbitrary yet reasonable, and to the best of our knowledge there is no obvious method for deciding how many genes to include. HOMER was run on FASTA mode using default parameters using the command ‘findMotifs.pl immune_promoter_seq.fasta homer_out -fasta other_promoter_seq.fasta’.

The output includes enriched known and *de novo* motifs. For the enriched known motifs, it found 42 motifs, with the top ones corresponding to the ETS family:

Total Target Sequences = 500, Total Background Sequences = 20218

Rank	Motif	Name	P-value	log P-value	q-value (Benjamini)	# Target Sequences with Motif	% of Targets Sequences with Motif	# Background Sequences with Motif	% of Background Sequences with Motif	Motif File	SVG
1		PU.1(ETS)/ThioMac-PU.1-ChIP-Seq(GSE21512)/Homer	1e-24	-5.735e+01	0.0000	223.0	44.60%	4664.4	23.07%	motif file (matrix)	svg
2		SpiB(ETS)/OCILY3-SPIB-ChIP-Seq(GSE56857)/Homer	1e-24	-5.530e+01	0.0000	142.0	28.40%	2284.9	11.30%	motif file (matrix)	svg
3		Ets1-distal(ETS)/CD44-PolII-ChIP-Seq(Barski_et_al)/Homer	1e-16	-3.813e+01	0.0000	142.0	28.40%	2775.7	13.73%	motif file (matrix)	svg
4		Etv2(ETS)/ES-ER71-ChIP-Seq(GSE59402)/Homer	1e-16	-3.700e+01	0.0000	305.0	61.00%	8557.9	42.33%	motif file (matrix)	svg
5		ETS1(ETS)/Jurkat-ETS1-ChIP-Seq(GSE17954)/Homer	1e-13	-3.075e+01	0.0000	322.0	64.40%	9605.8	47.51%	motif file (matrix)	svg
6		ERG(ETS)/VCaP-ERG-ChIP-Seq(GSE14097)/Homer	1e-12	-2.857e+01	0.0000	389.0	77.80%	12657.4	62.60%	motif file (matrix)	svg
7		Fli1(ETS)/CD8-FLI-ChIP-Seq(GSE20898)/Homer	1e-12	-2.771e+01	0.0000	339.0	67.80%	10507.2	51.97%	motif file (matrix)	svg
8		EWS-ERG-fusion(ETS)/CADO_ES1-EWS-ERG-ChIP-Seq(SRA014231)/Homer	1e-11	-2.640e+01	0.0000	215.0	43.00%	5727.6	28.33%	motif file (matrix)	svg
9		PU.1-IRF8(ETS-IRF)/pDC-Irf8-ChIP-Seq(GSE66899)/Homer	1e-10	-2.439e+01	0.0000	88.0	17.60%	1662.1	8.22%	motif file (matrix)	svg
10		Elf4(ETS)/BMDM-Elf4-ChIP-Seq(GSE8699)/Homer	1e-10	-2.387e+01	0.0000	313.0	62.60%	9678.7	47.87%	motif file (matrix)	svg
11		IRF8(IRF)/BMDM-IRF8-ChIP-Seq(GSE77884)/Homer	1e-8	-1.994e+01	0.0000	121.0	24.20%	2849.4	14.09%	motif file (matrix)	svg

Fig. R5. Top 11 enriched motifs in promoters of highly expressed immune cell genes as found by HOMER.

This result is consistent with our findings which suggested that ETS motifs are important in immune cells (Fig 4a). However, it is not straightforward to compare the results further as HOMER only identifies the motif enrichment and not the contribution to gene expression.

We also considered the *de novo* motif analysis, where the top hit is also an ETS motif. The remaining motifs are more difficult to interpret, but they are all at low frequencies, suggesting that they are unlikely to have a widespread impact on gene expression.

Rank	Motif	P-value	log P-value	% of Targets	% of Background	STD(Bg STD)	Best Match/Details
1		1e-30	-7.004e+01	49.60%	25.11%	225.8bp (325.8bp)	PB0058.1_Sfpi1.1/Jaspar(0.940) More Information Similar Motifs Found
2		1e-13	-3.051e+01	2.00%	0.02%	216.6bp (282.5bp)	Unknown2/Drosophila-Promoters/Homer(0.674) More Information Similar Motifs Found
3		1e-12	-2.942e+01	4.40%	0.50%	233.5bp (360.8bp)	Hth/dmmpmm(Noyes_hd)/fly(0.657) More Information Similar Motifs Found
4		1e-12	-2.850e+01	22.20%	10.81%	278.3bp (346.5bp)	EWS-ERG-fusion(ETS)/CADO_ES1-EWS-ERG-ChIP-Seq(SRA014231)/Homer(0.735) More Information Similar Motifs Found
5		1e-12	-2.845e+01	3.20%	0.22%	260.4bp (273.7bp)	CG17838(RRM)/Drosophila_melanogaster-RNCMPT00131-PBM/HughesRNA(0.695) More Information Similar Motifs Found
6 *		1e-11	-2.663e+01	2.20%	0.07%	236.4bp (374.0bp)	PB0194.1_Zbtb12.2/Jaspar(0.655) More Information Similar Motifs Found
7 *		1e-11	-2.661e+01	8.60%	2.38%	285.6bp (294.4bp)	HES6/MA1493.1/Jaspar(0.767) More Information Similar Motifs Found
8 *		1e-11	-2.554e+01	2.20%	0.07%	320.2bp (427.1bp)	vi/MA1462.1/Jaspar(0.652) More Information Similar Motifs Found
9 *		1e-10	-2.517e+01	5.00%	0.85%	241.2bp (348.9bp)	RUNX-AML(Runt)/CD4+-PolII-ChIP-Seq(Barski_et_al)/Homer(0.782) More Information Similar Motifs Found
10 *		1e-10	-2.473e+01	4.60%	0.72%	316.4bp (325.8bp)	ERF15(AP2EREBP)/colamp-ERF15-DAP-Seq(GSE60143)/Homer(0.690) More Information Similar Motifs Found
11 *		1e-10	-2.395e+01	2.60%	0.17%	316.3bp (250.2bp)	RME1/MA0370.1/Jaspar(0.698) More Information Similar Motifs Found
12 *		1e-10	-2.378e+01	4.20%	0.61%	192.5bp (325.0bp)	TFAP2A/MA0003.4/Jaspar(0.770) More Information Similar Motifs Found

Fig. R6. Top 12 *de novo* motifs in promoters of highly expressed immune cell genes as found by HOMER.

We conclude that the motifs identified by *cover* and HOMER are qualitatively similar.

We also compared against FIMO for the purpose of expression prediction and there we also found that overall, our integrated approach of identifying convolutional filters and their weights across pools was better suited for prediction of expression (see point below).

13. Have they considered showing the results of a simple approach like motif-matching using FIMO on the DNA sequences and regressing the motifs against the observed expression? The weights of this regression problem can be considered as motif influence scores.

We thank the reviewer for this suggestion, but we note that it is not altogether straightforward to implement since one needs to find a threshold to determine which motifs to include.

Nevertheless, we adopted an approach where we ran FIMO using the database of *all* human motifs from CIS-BP for the human kidney and human brain datasets, and using the database of mouse motifs from CIS-BP for the Tabula muris dataset. The input sequences for FIMO were

exactly the same as the inputs to the neural network, but in fasta form. For each case, we used the FIMO default options with the command 'fimo -o output_directory database.meme input_sequences.fa'. This generated a directory 'output_directory' with match scores for each database motif in each sequence. We loaded the match scores into an array 'X_train' of size num_sequences by num_motifs, with a default score of zero in case no motif match was found. We split the datasets up into the exact same train and test sets as the ones used by scover, and used python package *scikit-learn* to implement a linear regression on this data using command 'reg = LinearRegression().fit(X_train, Y_train)' where Y_train is the same data matrix that was predicted by scover as the target. We predicted on the test set using the command 'Y_pred = reg.predict(X_test)' and calculated the Pearson correlation between Y_pred and Y_test. We repeated this for each of the 10 folds of the data and used this to compare to the predictions made by scover. The workflow was the same for the random forest models, but instead of a linear regression model, the random forest was implemented using *scikit-learn* using command 'reg = RandomForestRegressor(n_estimators=20).fit(X_train, Y_train)'. This instantiates and trains a random forest regression with 20 estimators.

The results demonstrate that scover provides a significantly better fit for both the human kidney data (**Fig R1a,b**, $p=0.001$ compared to linear regression, $p=2.9e-06$ compared to random forest) and the Tabula Muris data ($p=0.008$ compared to linear regression, $p=1.9e-05$ compared to random forest). These benchmarks demonstrate that scover's approach of integrating motif discovery and regression improves prediction of expression compared to the alternative strategy of carrying out the two tasks separately. We expect that this is partly due to the nonlinearity following the convolutional layer; this allows the model to adapt the convolutional filter weights to adjust the threshold for when a motif match is close enough to be above zero. In contrast, a direct linear regression on FIMO match scores does not allow for changing the motif match threshold.

However, this result did not hold for the predictions on the human brain dataset, where a linear regression on the FIMO motif output outperformed scover (**Fig R1c**, $p = 9.82e-11$).

14. It seems that there are cases where a TF has a high influence score, but none of the TFs in the corresponding family is expressed. Can the authors explain in detail why this might happen? Can they show a plot of Expression (y-axis) and Influence-score (x-axis)?

We are somewhat confused by the reviewer's question. Figures 3c and 5d already contain some of the requested plots of expression vs influence scores. Moreover, based on our interpretation of Figs 3b and 5c, there are no clear-cut examples of cases where a motif has a high influence score, but all corresponding TFs are lowly expressed. We recognize that this may be a matter of interpretation, however, and what we consider reasonably expressed (e.g. NRF1 in the kidney) is lowly expressed by the reviewer's definition. We agree that the scenario described by the reviewer is problematic for our model, but we are unable to see any strong evidence that it has materialized.

15. Is the influence score matrix processed by normalizing? Can they show a PCA of individual cells with just the input expression values, i.e. PCA of input data to get an understanding of how much improvement in clustering is obtained by the model?

We apologize for the lack of clarity here. Typically, the scale of the influence scores varies quite a bit between the motif families. This is why we decided to z-normalize the influence score matrices in some visualizations, for instance in Fig 2a and 4a. The z-transformation helps to highlight differences across cell types for a given motif. For some of the figures, the raw scores are shown, for instance in Fig 6d. For any of the analyses in which we correlate the influence scores to other values, we do not normalize the scores.

As the reviewer alludes to, we expect that the input expression values should give a similar clustering as our model. In fact, the main goal of our method is not to cluster the data, and we view the clustering results as a validation that the model is able to identify cell type specific signals. This is confirmed by the PCA plots in **Fig R7** which represent a low dimensional projection of the expression data. Comparison to Figs 2c and 4c reveals that they are qualitatively similar. We do not expect that the model finds a better representation of the data, but we use the PCA to see how much the motif-by-pool influence score matrix can find the variation between cells. If not much variability can be explained by the motifs alone, we expect that the cell pools do not separate well.

Figure R7: (a) PCA of the expression levels for the cell pools from the kidney dataset. Colors are the same as Fig 2c in the main text. (b) PCA of the expression levels for the cell pools from the Tabula Muris dataset. Colors are the same as Fig 4b in the main text. (c) PCA of the pooled ATAC-seq levels for the cell pools from the human brain dataset. Colors are the same as Fig 6c in the main text.

16. Citations: Some relevant papers to cite

Fang, R. et al. Comprehensive analysis of single cell ATAC-seq data with SnapATAC. Nature Communications 12, 1337 (2021)

Kshirsagar M, et al. BindVAE: Dirichlet variational autoencoders for de novo motif discovery from accessible chromatin. Genome biology (2022)

Haghverdi, L., Lun, A.T.L., Morgan, M.D. & Marioni, J.C. Batch effects in single-cell RNA-sequencing data are corrected by matching mutual nearest neighbors. Nat Biotechnol

We thank the reviewer for their suggestions of citations and we have included the first two in the updated manuscript.

17. Have the authors considered incorporating longer-range dependence into the model? For instance, using dilated convolutions? Would this be helpful?

We are certain that the predictions could be better if we took into account further sequence for the prediction, since we do not take into account anything beyond 500 base pairs either way from the TSS, and longer-range dependencies are known to be important for tuning expression levels of genes. This worked well for e.g. Basenji (16) and Enformer (17). However, this would likely require more training data; especially in the case of the promoter model, since the set of promoter regions is fixed and limited to tens of thousands of sequences. Nevertheless, this worked for Enformer since they predict a multi-dimensional signal associated with each 128bp bin, and as such, they have more input data + target signal pairs.

In addition, if we wanted to reach the wider sequence we would indeed have to include a number of dilated convolutions, and this would hamper the interpretability of the model as motif representations could be distributed, *i.e.* they could be assembled using combinations of convolutions from the first and second layers. Then, we would not be able to find these whole motif representations that are linked to a single convolutional filter. In such a case, to find motifs it would be possible to use saliency/attribution methods like input times gradient or TF-MoDISco (18) to see which nucleotide changes would change the output signal the most, but it would be hard to then perturb these in a way other than a type of saturation mutagenesis to find out their influence scores. All in all, there are benefits and downsides to using either; we chose for simplicity here.

Minor:

In choosing the number of pools, is the hope that the number of pools == number of cell types? What about overlapping pools, where a cell is present in several pools? And what about cells from a single cell type being scattered across several different homogenous pools? Or worse, heterogeneous pools?

The reviewer raises several interesting points regarding the choice of pool size. In general, we do not expect the number of pools to equal the number of cell types since the abundance of cell types typically differs while the number of cells per pool is fixed. Thus, a better rule of thumb would be to have $\text{minimum}(\text{cell type size}) = \text{pool size}$ to ensure that each cell type has the potential to have at least one pool.

We have investigated the overlap between pools, and we find that for each individual pool, the fraction of non-uniquely assigned cells in the pool is generally high (**Fig. S3d**). However, this is to be expected for higher pool sizes: a larger number of pools will make the pooling operation like

a more densely sampled running average. In addition, we do not expect this to be a problem, since for most pools, the largest cell type fraction in the pool is >80% (Fig. S3a).

Figure S3: Statistics of the different datasets after pooling. (a) Largest cell type fraction per pool after pooling for the different datasets. (b) Number of cell types per pool for the different pooled datasets. (c) Total times a cell is included in pools for cells in the different datasets. (d) Fraction of non-uniquely assigned cells in a pool for the different pooled datasets.

Since pools are typically smaller than cell types, cells from the same cell type will by necessity be spread across multiple pools. However, as our analysis shows, most pools are homogenous, so this is unlikely to be problematic in practice. We do, however, believe that heterogeneous pools are valid and can serve a useful purpose if there is a continuous gradient present (e.g. a differentiation trajectory). In this scenario, hard clustering imposes an arbitrary cutoff which the pooling approach overcomes.

Page-7, Line-35: summing over all motifs in a cluster will usually be a large positive value. Also, summing over all pools will have the same issue. Are there any cases where this value is a small number? Does averaging make more sense?

The reviewer is correct in that this can result in a large number, either positive or negative. We can of course imagine strange scenarios where this could be problematic, but in practice we have not observed any of this so far. We do not think that averaging would help as it would just scale all of the numbers by the same factor, but we appreciate that we may have misunderstood the reviewer here. Since we mainly use the summed motif influence scores for downstream correlation analyses, we do not expect that the scale of the result matters.

What is the distance measure used to compute similarity between the cells by the Pooling/sketching algorithm? Was UMAP dimensionality reduction used to get the embeddings for the single cells?

We apologize for the lack of clarity. The k -nearest neighbors between the cells by the pooling/sketching algorithm were calculated using the PCA space where the first PC was removed since this was mainly associated with the number of counts in the cells. This k -nearest neighbor space was then used to calculate the UMAP embedding. It was across this UMAP embedding that we evenly sampled a number of seed cells using geometric sketching, but afterwards the k -nearest neighbors were used to select and pool together similar cells.

How was the number of 1000 seed cells for the human kidney dataset decided or 100 cells for the P0 dataset? Was it based on the size of the dataset (number of cells sequenced), or depth/quality of the reads?

We apologize for the lack of clarity. The number of seed cells was chosen as a trade-off between model size and capturing the full variability in the dataset. The larger the number of seed cells, the more of the phenotypic space we cover. Since the final layer of the neural network has $(n_{\text{motifs}} + 1) * n_{\text{seed_cells}}$ weights, the number of pools chosen directly scales the number of parameters by $(n_{\text{motifs}} + 1) * \text{additional_pools}$. In the new version of the manuscript we have chosen 1000 seed cells for each of the datasets for consistency.

Is the sampling of cells going to be sensitive to experimental variation - for ex: cells from the same sequencing depth will cluster together. How do they address other such factors?

The reviewer is correct that cells with the same sequencing depth could cluster together. To mitigate this, we calculated the k -nearest neighbors using the PCA space where the first principal component was removed. This was because the first principal component was associated with sequencing depth, as shown below for the human kidney dataset. As such, we expect that when we sample the seed cells using geometric sketching along the UMAP space (which is based on the k -nearest neighbor connectivity graph) it will be less based on the sequencing depth. Furthermore, the k -nearest neighbors are also used for the pooling operation itself. However, we cannot ensure that variability in the sequencing depth along the pools remains, but we do not assume that this is purely a technical artifact; some cell types will likely naturally include higher transcript counts.

Fig R8. PCA plot for the human kidney dataset before pooling. The color scale shows the number of counts in each cell.

It would be good to explain the biological motivation behind the bias term - what other factors that are not TFs influence the outcome?

The reviewer raises an interesting point. Our model only accounts for short motifs and although regulation through TF binding has been widely studied, features such DNA methylation, histone modifications, as well as factors influencing mRNA degradation also influence expression levels (to name a few). Hence, it is difficult to ascribe the bias term to a single factor, but it is rather “everything influencing expression levels that is not motifs”. Typically, the bias term is relatively small compared to the coefficients, so we expect there to be little need to explain it since it does not influence the outcome much.

On the human kidney dataset, it is not clear what the plots in Fig 3(a) are depicting. It seems like most promoter regions that have the motif being shown are randomly distributed in the plot?

In the UMAP, each point represents a promoter region and each promoter is associated with a vector corresponding to the output of the 6,000 convolutional filters. Hence, the map represents the similarity of the promoters as defined by their regulatory motif scores. Some of the motifs do not seem to exhibit any particular organization, while others show clear structure. For example, E2F and NRF1 are found in distinct regions of the UMAP, suggesting that the genes regulated by these factors share regulatory similarity. By contrast, ZFX is diffuse, suggesting that this motif is used by a wider range of promoters.

The observed values "t_obs" - is it the average of values from a pool?

It is a vector across the pools, containing the pooled transcriptomic or chromatin accessibility values (we do not consider the values before pooling during training).

Why did they use 10 CNN models - what variability does each represent?

As scover is based on a stochastic optimization method, results may vary between runs. We wanted to ensure that results are robust with regards to such variability, so that is why the same model is run multiple times. The choice of 10 models is empirical, and it is a trade-off between robustness and computational costs. We believe that 10 models is sufficient to identify results that are difficult to reproduce while keeping the run times reasonably low.

"...biggest difference in influence correlated...". Please mention the difference in influence of?

With the new ELU model, this is no longer true, so the statement has been removed from the new version of the manuscript.

Bibliography

1. Hie B, Cho H, DeMeo B, Bryson B, Berger B. Geometric Sketching Compactly Summarizes the Single-Cell Transcriptomic Landscape. *Cell Syst.* 2019 Jun 26;8(6):483-493.e7.
2. Trevino AE, Müller F, Andersen J, Sundaram L, Kathiria A, Shcherbina A, et al. Chromatin and gene-regulatory dynamics of the developing human cerebral cortex at single-cell resolution. *Cell.* 2021 Sep 16;184(19):5053-5069.e23.
3. Buske FA, Bodén M, Bauer DC, Bailey TL. Assigning roles to DNA regulatory motifs using comparative genomics. *Bioinformatics.* 2010 Apr 1;26(7):860–6.
4. Kim J, Cunningham R, James B, Wyder S, Gibson JD, Niehuis O, et al. Functional characterization of transcription factor motifs using cross-species comparison across large evolutionary distances. *PLoS Comput Biol.* 2010 Jan 29;6(1):e1000652.
5. Chandrasekaran S, Ament SA, Eddy JA, Rodriguez-Zas SL, Schatz BR, Price ND, et al. Behavior-specific changes in transcriptional modules lead to distinct and predictable neurogenomic states. *Proc Natl Acad Sci USA.* 2011 Nov 1;108(44):18020–5.
6. Whitney O, Pfenning AR, Howard JT, Blatti CA, Liu F, Ward JM, et al. Core and region-enriched networks of behaviorally regulated genes and the singing genome. *Science.* 2014 Dec 12;346(6215):1256780.
7. Dotson GA, Chen C, Lindsly S, Cicalo A, Dilworth S, Ryan C, et al. Deciphering multi-way interactions in the human genome. *Nat Commun.* 2022 Sep 20;13(1):5498.
8. Vierstra J, Lazar J, Sandstrom R, Halow J, Lee K, Bates D, et al. Global reference mapping of human transcription factor footprints. *Nature.* 2020 Jul 29;583(7818):729–36.
9. Khan A, Fornes O, Stigliani A, Gheorghe M, Castro-Mondragon JA, van der Lee R, et al. JASPAR 2018: update of the open-access database of transcription factor binding profiles and its web framework. *Nucleic Acids Res.* 2018 Jan 4;46(D1):D260–6.
10. Koo PK, Eddy SR. Representation learning of genomic sequence motifs with convolutional neural networks. *PLoS Comput Biol.* 2019 Dec 19;15(12):e1007560.
11. Yan J, Qiu Y, Ribeiro Dos Santos AM, Yin Y, Li YE, Vinckier N, et al. Systematic analysis

- of binding of transcription factors to noncoding variants. *Nature*. 2021 Mar;591(7848):147–51.
12. Lubana ES, Trivedi P, Hougen C, Dick RP, Hero AO. OrthoReg: Robust Network Pruning Using Orthonormality Regularization. *arXiv*. 2020;
 13. Ayinde BO, Inanc T, Zurada JM. Regularizing deep neural networks by enhancing diversity in feature extraction. *IEEE Trans Neural Netw Learn Syst*. 2019 Sep;30(9):2650–61.
 14. Zhang H, Lu T, Liu S, Yang J, Sun G, Cheng T, et al. Comprehensive understanding of Tn5 insertion preference improves transcription regulatory element identification. *NAR Genom Bioinform*. 2021 Dec;3(4):lqab094.
 15. Hu MC, Mo R, Bhella S, Wilson CW, Chuang P-T, Hui C-C, et al. GLI3-dependent transcriptional repression of Gli1, Gli2 and kidney patterning genes disrupts renal morphogenesis. *Development*. 2006 Feb;133(3):569–78.
 16. Kelley DR, Reshef YA, Bileschi M, Belanger D, McLean CY, Snoek J. Sequential regulatory activity prediction across chromosomes with convolutional neural networks. *Genome Res*. 2018 May;28(5):739–50.
 17. Avsec Ž, Agarwal V, Visentin D, Ledsam JR, Grabska-Barwinska A, Taylor KR, et al. Effective gene expression prediction from sequence by integrating long-range interactions. *Nat Methods*. 2021 Oct 4;18(10):1196–203.
 18. Shrikumar A, Tian K, Avsec Ž, Shcherbina A, Banerjee A, Sharmin M, et al. Technical Note on Transcription Factor Motif Discovery from Importance Scores (TF-MoDISco) version 0.5.6.5. *arXiv*. 2018;

Second round of review

Reviewer 2

This work has improved greatly since the main submission thanks to the addition of further experiments that illustrate the strength of this approach. Below are some remaining concerns, which would be ideal to address going forward.

Major concerns:

- If the goal is to predict expression or accessibility from sequence, then this work falls into the space dominated by approaches like BPNet and Enformer. How does the contribution from this work differ from those in terms of the problem setting, advantages and disadvantages. Please add analysis / experiments as needed to support your justification.
- Highlight clearly the biological settings in which this method is useful, in the Introduction? Please include a discussion of the novelty

Other concerns:

- Since the goal of the first preprocessing step: pooling is to get cells into homogenous groups, can other works from the literature that cluster scRNAseq data be more beneficial to use?
- I am not convinced by the multimodal setup in that it is not strictly multimodal. The accessibility data is mainly used to learn the model. The expression data is used to inform the pooling. Is there a big difference in the pools that are obtained by using just the accessibility or the expression alone?
- What are the advantages of the multimodal analysis? In the data on developing human cerebral cortex, it seemed like joint analysis (including both expression and accessibility) was explaining lesser of the observed variance?
- Ensure that all steps needed to replicate the results are described in detail
- How is PCA run, how UMAP was run: command used, distance metric, choice of algorithm to get the embedding
- Share the following data: raw reads, processed matrices from ATAC-seq / RNA-seq data, pooled datasets
- Share the models: neural network trained models for all experiments
- Share the detailed implementation: code used to go from processed data matrices, generate pooled data, run neural network training, motif generation from convolutional filters, scripts used to analyze the model and results, scripts that show how to run the code (it is OK if the code is not very clean as long as it can be run)

We would like to thank the reviewer for their constructive comments and the editor for allowing us to resubmit our manuscript. We have addressed all the comments made by the reviewer, and we believe this has strengthened our manuscript significantly.

All changes made to the manuscript have been highlighted in red.

Reviewer #1: Major concerns:

- If the goal is to predict expression or accessibility from sequence, then this work falls into the space dominated by approaches like BPNet and Enformer. How does the contribution from this work differ from those in terms of the problem setting, advantages and disadvantages. Please add analysis / experiments as needed to support your justification.

We thank the reviewer for this suggestion and we are happy to clarify. Our method differs from methods such as BPNet, Enformer, and Basset in a few ways. The other methods take as input a sequence and multiple signals (e.g. CHIP-seq or ATAC-seq) across that stretch of nucleotides, and they then learn to predict the signals. In particular, BPNet predicts single base pair resolution CHIP-nexus data from sequence, and Enformer predicts functional signals in 128bp bins across a large sequence. In contrast, we aim to predict a single number that is associated with the sequence. In the case of predicting scRNA-seq data, we predict a single expression value (in each cell) for a given promoter, whereas in the open chromatin case, we predict a single accessibility value for each accessible region.

Method	Predicts	Model architecture	Input sequence length
Xpresso	Steady-state mRNA levels	CNN with multiple convolutional layers	10,500 bp
scBasset	scATAC-seq data	CNN with multiple convolutional layers	1,344 bp
Enformer	Genome-wide CAGE-seq and epigenetic signals	CNN with multiple convolutional & transformer layers	196,608 bp
Sei	Genome-wide epigenetic signals	CNN with multiple convolutional layers	4,096 bp
BPNet	Genome-wide base-resolution CHIP-nexus signal	CNN with multiple convolutional layers	1,000 bp
Scover (this study)	Pooled scRNA-seq or pooled scATAC-seq data	CNN with one convolutional layer	1,000 bp (expression) or 240 bp (accessibility)

Table R1. Comparison of deep learning approaches for regulatory sequence signal prediction. Included as Table S1 in the new version of the manuscript.

The other methods are typically designed to predict the associated signals for the entire genome. Instead, we predict single values for a subset of the genome containing 1,000 bp sequences (in the case of expression prediction) or 240 bp sequences (in the case of accessibility prediction). Consequently, the training dataset is of a smaller size when considering chromatin accessibility. For gene expression data this is typically not a problem since the number of genes is fixed and independent of the granularity. Another important difference is that scover can operate on single cell data, whereas the CAGE-seq data

predicted by Enformer is less widely used, and to the best of our knowledge the CAGE protocol cannot produce single cell data.

The method that is most similar to ours is scBasset [Yuan and Kelley, *Nat Methods* 2022]. scBasset predicts the accessibility of a 1,344 bp sequence in multiple cells using a multi-layer convolutional neural network. As such, the analysis is slightly different, since the convolutional filters cannot easily be linked to transcription factor database motifs, and instead scBasset learns a distributed representation of motifs. Furthermore, scBasset was not designed to predict gene expression. Another similar method is the Xpresso model [Agarwal and Shendure, *Cell reports* 2020], which predicts gene expression from promoter sequences. However, Xpresso predictions are for the same cell type that it was trained on, and it also contains multiple convolutional layers. Although we are aware of a considerable interest within the field for developing approaches to make multi-layer representations of sequence binding motifs more easily interpretable, we argue that there is value to a single-layer approach where interpretation is trivial. Our method is the only method we are aware of that predicts gene expression in single cells while simultaneously identifying regulatory motifs.

We have extended the introduction to highlight these differences to readers.

Highlight clearly the biological settings in which this method is useful, in the Introduction?
Please include a discussion of the novelty

We thank the reviewer for the suggestion. To address this, we have now rewritten the introduction to discuss the novelty of the method and the biological settings where it can be useful. As discussed above (and in the previous rebuttal letter), our main novelty is that scover simultaneously identifies motifs and their impact on a related signal, along with the ease of interpretation of the representation. The situations where we think scover will be most useful include:

- We think that the most important use case for scover will be as a hypothesis generator. In a scenario where one is interested in narrowing down candidate motifs identified from a single cell experiment to carry out low-throughput validation experiments, scover will be helpful in prioritizing what motifs/TFs to investigate.
- Related to the above, scover allows the user to quantitatively compare the contributions of different motifs to gene expression/open chromatin.
- We also expect our method to be useful in non-model organisms for which databases of TFs and motifs are not as complete. If the genomic sequence, gene locations, and single-cell datasets are available, our method can identify the main predictive sequence motifs and how their influence varies across cell pools.

Other concerns:

- Since the goal of the first preprocessing step: pooling is to get cells into homogenous groups, can other works from the literature that cluster scRNAseq data be more beneficial to use?

The reviewer raises an important question. We have now compared two other ways to sample cells, in addition to geometric sketching:

- SEACells

- Random selection of cells

Firstly, we compared our approach to the recently published SEACells [Persad et al, *Nat Biotech* 2023]. SEACells was designed to find combinations of cells to form cell pools that retain biological variation to improve the chances of including rare cell populations as a single group. We applied SEACells to the human kidney dataset using the default parameters. To compare SEACells to our sketching approach, we considered the metrics used in Fig S1. The figure is reproduced below along with the corresponding results for SEACells (please note that **Fig R1d** represents a different metric to **Fig S1d**). Both in terms of the largest cell type fraction in a given pool, and in terms of the number of cell types in a pool, the performance of the two approaches is similar (**Fig R1a,b**). The high degree of similarity strongly suggests that the downstream results will not differ substantially. One difference between our approach and SEACells is that SEACells makes sure every cell is only included once in the final dataset (**Fig R1c**), which means that the metacells created by SEACells can be of different sizes (**Fig R1d**). As a result, it is necessary to normalize the data afterwards to get the same number of total counts in each cell, and this might require some changes in how the model is trained, but we have not yet worked out the details.

We also applied SEACells to the Tabula Muris and human brain datasets using the default parameters (**Fig R2, 3**). We applied it to the RNA modality of the multimodal dataset for fair comparison, since our pooling approach also made use of the representation calculated from the RNA modality of the dataset. SEACells slightly outperformed our approach in terms of the cell type purity (**Fig R2a, R3a**) and the number of cell types in each pool (**Fig R2b, R3b**). However, the higher purity is most likely also driven by the smaller number of cells per pool for SEACells.

We have updated the text to mention the possibility of using SEACells for pooling cells, but we have not updated the results in the manuscript. We've further included a jupyter notebook on our github repository that shows how to use SEACells to generate a dataset that can be used with our method.

Figure R1. Statistics of the human kidney datasets made using the original or SEACells approach. (a) Largest cell type fraction per pool after pooling for the different methods. (b) Number of cell types per pool for the different methods. (c) Total times a cell is included in pools for cells in the different methods. For SEACells, all the values are 1. (d) Total number of cells in a pool for the different methods. For Original, all the values are 120.

Figure R2. Statistics of the Tabula Muris datasets made using the original or SEACells approach. (a) Largest cell type fraction per pool after pooling for the different methods. (b) Number of cell types per pool for the different methods. (c) Total times a cell is included in pools for cells in the different methods. For SEACells, all the values are 1. (d) Total number of cells in a pool for the different methods. For Original, all the values are 80.

Figure R3. Statistics of the human brain datasets made using the original or SEACells approach. (a) Largest cell type fraction per pool after pooling for the different methods. (b) Number of cell types per pool for the different methods. (c) Total times a cell is included in pools for cells in the different methods. For SEACells, all the values are 1. (d) Total number of cells in a pool for the different methods. For Original, all the values are 100.

Previously, we used a different pooling strategy whereby pools were created by random sampling of cells within a cell type. The main advantage of this method is its simplicity - it is very easy to understand what is going on and it runs very fast. Next to this, the cell type purity in a given pool is always 100%. The main drawback is that many of the pools made for a given cell type will approximately represent the cell type average, since the approach does not take into account cell similarity. For these reasons, although results were still favorable with good R² values, it was abandoned in favor of the sketching approach.

- I am not convinced by the multimodal setup in that it is not strictly multimodal. The accessibility data is mainly used to learn the model. The expression data is used to inform the pooling. Is there a big difference in the pools that are obtained by using just the accessibility or the expression alone?

We thank the reviewer for this important question, and as suggested we have compared the pools generated using representations from the two modalities. Since the process to run UMAP on single-cell ATAC-seq data is slightly different to the process to run UMAP on single-cell RNA-seq, we expect the pooling to produce different results. In particular, the nearest neighbors (that are used for UMAP calculation) are calculated on the output of latent semantic indexing analysis instead of the output of the PCA analysis. Similarly to the comparison to SEACells, we compared the datasets generated using the RNA-seq and ATAC-seq representations using the dataset statistics from Figure S1 (**Fig. R4**). As can be seen, the pooled dataset generated with the ATAC representation has a lower cell type purity compared to the dataset generated by the RNA representation (**Fig. R4a**). This could potentially be the result of the cell type labels being determined by the RNA representation. Similarly, the number of cell types in the pools was higher when using the ATAC

representation (**Fig R4b**). The datasets are highly similar in terms of the total times the cells are included and the fraction of non-uniquely assigned cells in the pools (**Fig R4c,d**). Overall, since the statistics are on par or worse when using the ATAC-seq representation, we decided to stick with our original approach for pooling.

Figure R4. Statistics of the different datasets made using pools constructed using either the RNA or the ATAC representation of the human brain dataset. (a) Largest cell type fraction per pool after pooling for the different methods. (b) Number of cell types per pool for the different methods. (c) Total times a cell is included in pools for cells in the different methods. (d) Fraction of non-uniquely assigned cells in a pool for the different methods.

While we appreciate that the training setup is not multimodal, we would like to emphasize that the multimodal aspect of the data allows our downstream transcription factor expression analysis. Without the expression data from the same cells, this analysis would not be possible. To make sure that other readers do not have the same reaction as the reviewer we have highlighted these issues in the discussion.

- What are the advantages of the multimodal analysis? In the data on developing human cerebral cortex, it seemed like joint analysis (including both expression and accessibility) was explaining lesser of the observed variance?

We conjecture that there are a few explanations for the reduced explained variance in the multimodal dataset. Primarily, since we are predicting a different data modality (chromatin accessibility versus gene expression) with different sequences (distal enhancers versus promoter sequences), we expect that the distributions underlying both the input and output of the model compared to the gene expression prediction task differ significantly. The poorer performance suggests that the model has a hard time overcoming these discrepancies, and that a more complex model or more training data is required to improve performance.

The primary advantage of the multimodal analysis is that it allows us to first predict enhancer accessibility, and then identify putative transcription factors that are associated with the observed patterns using the transcriptional modality, since we have that same data from the same cells/pools.

- Ensure that all steps needed to replicate the results are described in detail
- How is PCA run, how UMAP was run: command used, distance metric, choice of algorithm to get the embedding

We appreciate the reviewer's suggestion. We have now included a more detailed description of the methods used in the Methods section, and we have shared details about the implementation of the code to process data and run the method (see below for the link to the github accompanying the manuscript).

The following python workflow was followed for constructing the datasets. For the human kidney dataset, the fetal and adult kidney datasets were concatenated using the `'concatenate()'` function of AnnData objects. Then, using the command `'sc.tl.pca(ad, svd_solver='arpack', n_comps=100)'` from Scanpy, the first 100 principal components were calculated. For all datasets, the first PC was not used for subsequent steps since it correlated highly with the number of counts in the pools. We removed it by running `'ad.obsm['X_pca'] = ad.obsm['X_pca'][:, 1:]'`. As the RNA datasets consist of concatenated datasets of different batches, batch-balanced k-nearest neighbor algorithm BBKNN was run with default parameters using command `'bbknn(ad, batch_key='dataset)'`. Subsequently, these k-nearest neighbors were used to calculate the UMAP representation using the Scanpy command `'sc.tl.umap(ad)'`. Using geometric sketching, a subset of 1,000 cells (the 'seed cells') was selected that spans the UMAP representation of the dataset uniformly. This was done using the geometric sketching function `'gs()'` from the 'geosketch' package: `'sketch_index = gs(ad.obsm['X_umap'], 1000, replace=False)'`. Then, k-nearest neighbors of these initial 'seed cells' were calculated again, but now *k* was set to be the pool size. For the implementation of this method, please refer to the function `'pool_anndata()'` in the github file `'scover/data/utils.py'` at <https://github.com/jacobhepkema/scover>. Briefly, k-nearest neighbors were calculated using Scanpy command `'sc.pp.neighbors(ad, n_pcs = 40, n_neighbors = k)'`, where *k* is the pool size. For each of the seed cells, the pooled counts were calculated as the sum of the raw counts for the seed cell and its *k*-nearest neighbors, followed by a $\log(1+\text{counts})$ transformation.

For the Tabula Muris dataset, the workflow was similar. 20 Tabula Muris FACS-sorted Smart-Seq2 datasets were concatenated in the cell dimension using the `'concatenate()'` function of AnnData objects. 297 erythrocytes were excluded from further analysis given their low counts. The datasets included were Trachea, Bladder, Heart, Limb_Muscle, Diaphragm, Fat, Lung, Aorta, Mammary_Gland, Brain_Non-Myeloid, Skin, Kidney, Liver, Pancreas, Tongue, Brain_Myeloid, Thymus, Spleen, Marrow, and Large_Intestine. PCA was calculated using the same command as for the human kidney dataset, and the same command was used to remove the first PC. BBKNN was run using a slightly different command: `'bbknn(ad, batch_key='dataset', trim=10000, approx=False, use_faiss=False)'`, since this command was also used by the BBKNN developers for their integration of Tabula Muris datasets. Then, the UMAP representation was calculated using Scanpy command `'sc.tl.umap(ad, min_dist=.3)'`. The seed cells were obtained using the same command as for

the human kidney dataset. The function `'pool_anndata()'` from our package was used with `'neighbors=80'` to create pools that aggregate 80 nearest neighbors each.

For the human brain dataset, we calculated the first 100 PCs for the RNA modality using the Scanpy command `'sc.tl.pca(ad_rna, svd_solver='arpack', n_comps=100)'`. Afterwards, the first PC was excluded using the command `'ad_rna.obsm['X_pca'] = ad_rna.obsm['X_pca'][:,1:]'`. Since there were not multiple datasets to be integrated, BBKNN was not applied. Instead, nearest neighbors were calculated using Scanpy command `'sc.pp.neighbors(ad_rna, n_neighbors=100, n_pcs=80)'`, and the UMAP representation was then calculated using Scanpy command `'sc.tl.umap(ad_rna)'`. The seed cells were obtained using the command `'sketch_index = gs(ad_rna.obsm['X_umap'], 1000, replace=False)'`. Next, the RNA dataset was pooled using the `'pool_anndata()'` command from our github package with argument `'neighbors=100'` to create pools that aggregate 100 nearest neighbors each. Then, the same cells that were used to pool the RNA modality were used to pool the ATAC modality of the dataset using the `'pool_anndata_given_pseudobulk_idx()'` function from our github: `'pool_anndata_given_pseudobulk_idx(ad_atac, pseudobulk_idx)'`.

- Share the following data: raw reads, processed matrices from ATAC-seq / RNA-seq data, pooled datasets

We have now included our pooled datasets in a Zenodo archive at <https://doi.org/10.5281/zenodo.8060659>. However, since we did not generate the raw data and processed matrices, we are not sharing these data ourselves. Instead, accession numbers are clearly listed in the manuscript.

- Share the models: neural network trained models for all experiments

We thank the reviewer for the helpful suggestion. We have now included the model parameters in the github repository.

- Share the detailed implementation: code used to go from processed data matrices, generate pooled data, run neural network training, motif generation from convolutional filters, scripts used to analyze the model and results, scripts that show how to run the code (it is OK if the code is not very clean as long as it can be run)

We thank the reviewer for the suggestion. We have now updated our scripts on our github to reflect the newest version of the model and the analysis. In particular, we would like to point the reviewer to the example jupyter notebooks included in the github: <https://github.com/jacobhepkema/scover>, in the directory `'example_notebooks'`.